# CompoundRay, an open-source tool for high-speed and high-fidelity rendering of compound eyes

**Blayze Millward\*, Steve Maddock, Michael Mangan**

Department of Computer Science, University of Sheffield, Sheffield, United Kingdom

**Abstract** Revealing the functioning of compound eyes is of interest to biologists and engineers alike who wish to understand how visually complex behaviours (e.g. detection, tracking, and navigation) arise in nature, and to abstract concepts to develop novel artificial sensory systems. A key investigative method is to replicate the sensory apparatus using artificial systems, allowing for investigation of the visual information that drives animal behaviour when exposed to environmental cues. To date, 'compound eye models' (CEMs) have largely explored features such as field of view and angular resolution, but the role of shape and overall structure have been largely overlooked due to modelling complexity. Modern real-time ray-tracing technologies are enabling the construction of a new generation of computationally fast, high-fidelity CEMs. This work introduces a new open-source CEM software (*CompoundRay*) that is capable of accurately rendering the visual perspective of bees (6000 individual ommatidia arranged on 2 realistic eye surfaces) at over 3000 frames per second. We show how the speed and accuracy facilitated by this software can be used to investigate pressing research questions (e.g. how low resolution compound eyes can localise small objects) using modern methods (e.g. machine learning-based information exploration).

## Editor's evaluation

In this important work, the authors develop compelling new open source methods to study compound eye vision, with particular emphasis and examples in insects and appropriately supporting arbitrarily diverse spatial distributions, types and mixtures of types of ommatidia. The manuscript introduces example experiments to illustrate the use of the new methodology. This work supports future studies of invertebrate brains, a timely addition to the newly mapped connectomes of insect brains.

**\*For correspondence:**
b.f.millward@sheffield.ac.uk

**Competing interest:** The authors declare that no competing interests exist.

## Introduction

Insects visually solve an array of complex problems including the detection and tracking of fast moving prey (*Wiederman et al., 2017*), long-distance navigation (*Wehner, 2020*), and even three-dimensional (3D) depth estimation (*Nityananda et al., 2018*). These capabilities are realised using a sensory apparatus that is fundamentally different from those of mammals. Therefore revealing the functional properties of the insect visual system offers insights for biologists as well as inspiration for engineers looking to develop novel artificial imaging systems (*Land and Fernald, 1992*; *Land, 1997*; *Arendt, 2003*; *Song et al., 2013*).

Arthropods possess two primary visual sensors known as compound eyes. Each eye is constructed from a patchwork of self-contained light-sensing structures known as ommatidia, each featuring a lens, a light guide, and a cluster of photosensitive cells (*Figure 1a*). Ommatidia are physically interlocked with their neighbours, together forming a bulbous outer structure (the compound eye itself) that can

**Figure 1.** The structure of the compound eye.

(a) A diagram of a single ommatidium, shown with the lensing apparatus at the top that guides light into the photo-sensitive cells below. (b) An image of a real compound eye consisting of hundreds of ommatidia, available in WikiMedia Commons copyright holder *Moussa Direct Ltd.*, copyright year 2001, distributed under the terms of a CC BY-SA 3.0 license (https://creativecommons.org/licenses/by-sa/3.0), and cropped with an added zoom indicator (red) from the original.

vary in size, shape, and curvature, offering a range of adaptations for particular tasks and environments (*Land and Nilsson, 2002*; *Figure 1b*). The properties of the lens and photo-receptors are fixed for individual ommatidia but can vary across regions of an individual's eye (*Meyer and Labhart, 1993*) as well as between individuals of different castes (*Collett and Land, 1975*) and species (*Land, 1989*). This arrangement of independent, interlocked light sensing elements and lenses differs greatly from the mammalian system that utilises a single lens to project a high-resolution image onto a retina.

Creation of compound eye models (CEMs), in both hardware or software, is a well-established mechanism to explore the information provided by compound eyes and assess their impact on behaviour. Insights derived from this methodology include demonstration of the benefits of a wide field of view (FOV) and low resolution for visual navigation (*Zeil et al., 2003*; *Vardy and Moller, 2005*; *Mangan and Webb, 2009*; *Wystrach et al., 2016*), and the role played by non-visible light sensing for place recognition (*Möller, 2002*; *Stone et al., 2006*; *Differt and Möller, 2016*) and direction sensing (*Lambrinos et al., 1997*; *Gkanias et al., 2019*). Yet, simulated CEMs tend to suffer from a common design shortcoming that limits their ability to accurately replicate insect vision. Specifically, despite differences in the sampling techniques used (e.g. see *Mangan and Webb, 2009*; *Baddeley et al., 2012* for custom sampling approaches, *Neumann, 2002*; *Basten and Mallot, 2010* for rendering based cubemapping CEMs, and *Polster et al., 2018* for ray-casting methods), all contemporary CEMs sample from a single viewpoint. In contrast, the distributed arrangement of ommatidia on distinct 3D

eye surfaces provides insects with a multi-viewpoint system that generates different information for different eye shapes.

To facilitate exploration of such features, e.g., the placement of ommatidia on arbitrary surfaces, an ideal rendering system would allow light to be sampled from different 3D locations through individually configured ommatidia replicating the complex structure of real compound eyes. High-fidelity compound-vision rendering engines with some of these features (though notably slower than real time) were developed previously (*Giger, 1996*; *Collins, 1998*; *Polster et al., 2018*) but were not widely adopted. Difficulties arising as a result of the computational complexity (and so execution time) of CEMs are diminishing as dedicated ray-casting hardware emerges that allows for the capture of visual data from multiple locations in parallel at high speed (e.g. Nvidia [Santa Clara, California, United States] *RTX* series GPUs [*Purcell et al., 2005*; *Burgess, 2020*]). Such systems present an ideal tool tferreo replicate insect vision in unprecedented accuracy at the speeds needed for effective exploration of the compound eye design space itself. As outlined in *Millward et al., 2020*, a next-generation insect eye renderer should:

1. Allow for the arrangement of an arbitrary number of ommatidia at arbitrary 3D points.
2. Allow for the configuration of individual ommatidial properties (e.g. lens acceptance angle).
3. Perform beyond real time to allow exploration of the design space.

This paper presents a ray-casting-based renderer, *CompoundRay*, that leverages modern hardware-accelerated ray-tracing (RT) graphics pipelines and fulfils all three of these criteria, allowing researchers in the fields of compound vision and biorobotics to quickly explore the impact varying eye designs have on an autonomous agent's ability to perceive the world.

## Materials and methods
### Ray-casting-based insect eye renderer

A common approach for general real-time rendering such as that found in video games and interactive displays follows polygon-projection-based methods inspired by the simple pinhole camera. As *Figure 2a* shows, these systems function by directly projecting the faces and vertices of the scene's 3D geometry through a singular imaging point and on to a projection plane. In contrast, compound visual systems essentially form multiple pinhole-like viewing systems (*Figure 2c*), each with their own image point. This is similar to the effects that mirrors, lenses, and other reflective, diffracting, and refractive

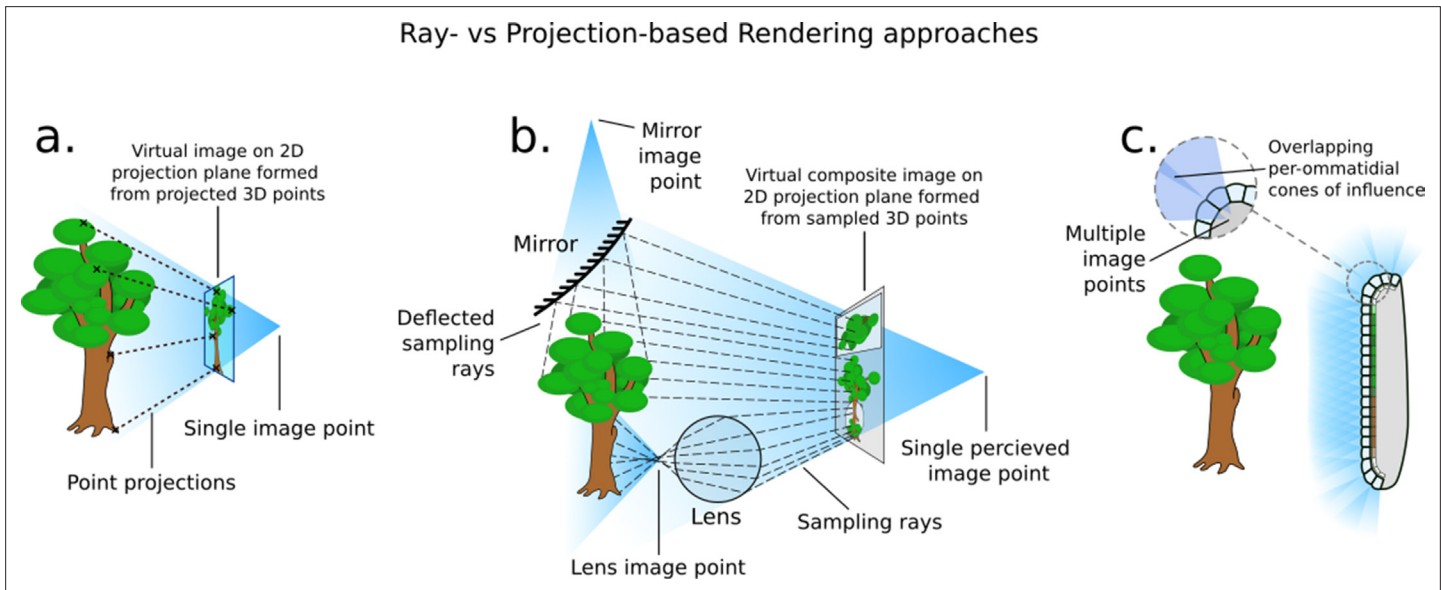

**Figure 2.** How different rendering approaches form differing image points (not to scale). (**a**) In three-dimensional (3D) projection-based rendering, the polygons that comprise the 3D geometry are projected down to a viewing plane, through one image point. (**b**) In 3D ray-casting-based rendering (used in this paper), rays are emitted into a scene and *sample* the 3D geometry they contact, potentially forming many image points. (**c**) A compound eye requires many image points in order to image its surroundings.

surfaces have on light transport, in which multiple imaging points are formed as the result of light being diverted from its original course (*Figure 2b*). A polygon projection approach will struggle with these optical phenomena as the process directly transforms the faces of the 3D object from the scene onto the camera's view plane, in effect applying a projection transform onto a plane or other regular surface forming a singular image point (*Figure 2a*).

Drawing on previous work from *Polster et al., 2018*, *CompoundRay* uses a ray-casting approach to rendering the insect visual perspective. Ray-based methods offer an alternative to projective transform rendering: rays are sent out from the virtual viewpoint, simulating – in reverse – the paths of photons from the scene into the cone of vision, accumulating colour from the surfaces they interact with (*Figure 2b*). This allows for surfaces such as mirrors and lenses to accurately reflect light being rendered. The term *ray-based methods* here is used as an umbrella term for all rendering approaches that primarily use the intersection of rays and a scene in order to generate an image. In particular, we refer to *ray casting* as the act of sampling a scene using a single ray (as per *Roth, 1982*) – a process that CompoundRay performs multiple times from each ommatidium – and *ray tracing* as the act of using multiple recursive ray casts to simulate light bouncing around a scene and off of objects within, similar to early work in the field of shading (*Appel, 1968*; *Whitted, 1979*). In this work we do *not* recursively cast rays, instead only using a single ray cast to sample directly the environment from an arbitrary point in 3D space (in this case, over the sampling cone of an ommatidium).

Ray-based approaches can be incredibly computationally complex, as each ray (of which there can be thousands per pixel) needs to be tested against every object in the scene (which can be composed of millions of objects) to detect and simulate interactions. Thus, they have historically only been used in offline cases where individual frames of an animation can take many hours to render, such as in cinema (*Christensen et al., 2018*), and prior to that in architectural and design drafting (*Appel, 1968*; *Roth, 1982*). In recent years, however, graphics processing units (GPUs) have been increasing in capability. In order to better capture the photo-realistic offerings of ray-based rendering methods, GPU manufacturers have introduced dedicated programmable RT hardware into their graphics pipelines (*Purcell et al., 2005*). These RT cores are optimised for efficient parallel triangle-line intersection, allowing billions of rays to be cast in real time into complex 3D scenes.

As compound eyes consist of a complex lensing structure formed over a non-uniform surface, they naturally form a multitude of imaging points across a surface (*Figure 2c*). These projection surfaces are often unique and varied, meaning it is practically infeasible to find a single projective transform to produce the appropriate composite view in a single projection-based operation. As a result, ray-based methods become the natural choice for simulating compound vision, as opposed to the more commonly seen projection-based methods. By utilising modern hardware, it is possible to effectively capture the optical features of the compound eye from the ommatidia up, in a high-fidelity yet performant way.

## Modelling individual ommatidia

As the compound eyes of insects contain many hundreds or even thousands of lenses (*Figure 1b*) and each acts to focus light to its own point (forming a unique perspective), ray-based methods become the natural technology to use to implement a simulator that can capture their unique sensory abilities. Here, a single ommatidium is first examined, simulated, and then integrated into a complete compound eye simulation.

Each individual ommatidium in a compound eye captures light from a cone of vision (defined by the *ommatidial acceptance angle*), much in the way that our own eyes observe only a forward cone of the world around us. However, in the case of the ommatidium, all light captured within the cone of vision is focused to a singular point on a small photo-receptor cluster, rather than the many millions of photo-receptors in the human eye that allow for the high-resolution image that we experience – in this way, the vision of a singular ommatidium is more akin to the singular averaged colour of all that is visible within its given cone of vision – its *sampling domain*, much as a pixel from a photo is the average colour of all that lies 'behind' it.

*CompoundRay* uses Monte Carlo integration (*Kajiya, 1986*) to estimate the light intensity from the scene within the ommatidium's sampling domain, effectively forming a small pinhole-type camera at the position of the ommatidium. These discrete samples must be accrued in alignment with the ommatidium's *sampling function* – as the lens gathers more light from forward angles than those

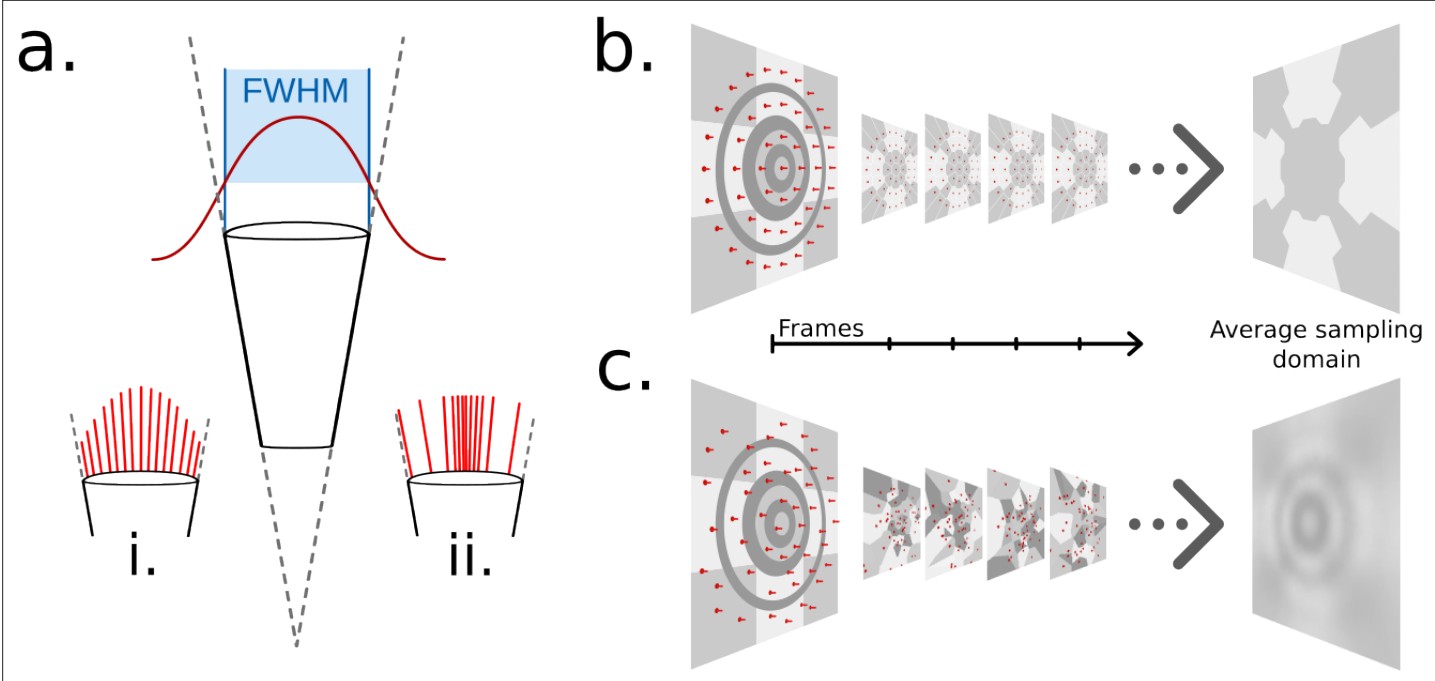

**Figure 3.** Ommatidial sampling distributions, statically and stochastically distributed. (**a**) The Gaussian sampling curve (red) with a full width at half maximum (FWHM) equal to the acceptance angle of each ommatidium. (**i**) Example statically distributed sampling rays, length indicates weight. (**ii**) Example stochastically distributed sampling rays, equal weight, distributed according to Gaussian distribution. (**b**) A concentric circular static sampling pattern (red dots) aliases away the three concentric dark grey rings present on a surface. (**c**) The per-sample colours as seen from a single ommatidium mapping to the samples in *b*, note the lack of any influence from the dark grey concentric rings, which have been aliased out.

diverging away from the ommatidial axis. Increasing the sample count will result in a more accurate estimate of the light converging on the point.

Previous works *Collins, 1998*; *Polster et al., 2018* have implemented this approximation of the light sampling function by assuming the influence of individual light rays, distributed statically across the sampling cone, is modulated dependent on the angle to the ommatidial axis via a Gaussian function with a full width at half maximum equal to the acceptance angle of the ommatidium, as seen in *Figure 3a*. *CompoundRay* also uses the Gaussian function as a basis for sampling modulation; however, unlike other works, it is not approximated via a static sampling pattern with weighting biasing the forward direction (as is seen in *Figure 3a:i*). Instead, sampling *direction* is modulated and chosen at random in relation to the Gaussian probability curve of the sampling function on a temporal, frame-by-frame basis. Each sample is assumed to have an equal weight, with the visual input to a single ommatidium then being formed as the average of all samples (*Figure 3a:ii*).

Static sampling can result in elements of the scene (in the case demonstrated in *Figure 3b*, dark grey circles) being aliased out of the final measurement of sensed light. Increasing per-ommatidium sampling rate *can* ease these aliasing problems (even in the case of a static sampling pattern) at close range but only serves to delay triggering aliasing until further distances where structural frequency again matches sampling pattern. Rather, by choosing to stochastically vary the sampled direction on a frame-by-frame basis removes the structured nature of any single frame's aliasing, as that same view point will generate different aliasing artefacts on the next frame, having the net effect of reducing structured aliasing when considering sequences of frames (*Figure 3c*). One benefit of this approach is that averaging renderings from the same perspective becomes synonymous with an increased sampling rate, indeed, this is how the renderer increases sampling rate internally: by increasing the rendering volume for the compound perspective, effectively batch-rendering and averaging frames from the same point in time. However, as each ommatidium will sample the simulated environment differently for every frame rendered, small variations in the colour received at any given ommatidium occur over time. This point is explored further in the 'Results' section.

In the natural compound eye, light focused from each ommatidium's sampling domain is transported via a crystalline cone and one or more rhabdomeres toward the receiving photo-receptor cluster – a process that has been modelled by *Song et al., 2009*. However, modelling the biophysical intricacies of the internal ommatidium structure is considered beyond the scope of this software and is considered a post-processing step that could be run on the generated image, considering each pixel intensity as a likelihood measure of photon arrival at an ommatidium.

## From single ommatidia to full compound eye

By arranging the simulated ommatidia within 3D space so as to mimic the way real compound eyes are arranged, an image can be generated that captures the full compound eye view. For this task, per-ommatidium facet diameter and acceptance angle can be used to generate the cone of influence of each ommatidium, and simulated ommatidia can be placed at any position and orientation within the environment. As the system is using RT techniques to effectively simulate pinhole cameras at each individual ommatidium, orientation and position relative to the lens of each ommatidium can be set with a single per-ommatidium data point (consisting of the position, orientation, and lens properties), allowing each ommatidium to spawn rays independently of each other. This has the benefit of allowing the system to simulate any type of eye surface shape, and even multiple eyes simultaneously, fulfilling the first of the three defined criteria: *allowing for the arrangement of an arbitrary number of ommatidia at arbitrary 3D points*. The generated images can then be used to allow human viewers to visualise the information captured by the eye by projecting this visual surface onto a two-dimensional (2D) plane using orientation-wise or position-wise spherical equirectangular Voronoi diagrams of each ommatidium's visual input. Alternatively, a simpler vectorised projection method that plots the visual inputs of each ommatidium in a vector can be used directly as a standardised input to computational models of insect neurobiology.

## The CompoundRay software pipeline

The renderer is written in C++ and the Nvidia Compute Unified Device Architecture (CUDA) GPU programming language and allows for a 3D environment representing a given experimental setup to be rendered from the perspective of an insect. The core of the renderer runs in parallel on the GPU (the *device*) as a series of CUDA shader programs, which are driven by a C++ program running on the host computer (the *host*). Environments are stored in *GL Transmission Format* (glTF; *Robinet et al., 2014*) files consisting of 3D objects and cameras, the latter of which can be of two types: compound (structured sets of ommatidia) or traditional (perspective, orthographic, and panoramic). Traditional cameras are implemented to aid the user in the design and analysis of their assembled 3D environment. Compound cameras contain all relevant information for rendering a view of the environment from the perspective of a compound eye. Each camera stores the information required for rendering its view (such as its orientation, position, FOV, or ommatidial structure) in an on-device data record data structure.

*Figure 4* shows the operational design of the renderer from the device side. The renderer is built on top of Nvidia's *Optix* (*Parker et al., 2010*) RT framework, which is a pipeline-based system: a pipeline is assembled, which then allows for parallel per-pixel rendering of the environment. A pipeline consists of a ray generator shader, a geometry acceleration structure (GAS), and numerous material shaders. A shader is a small program that is designed to be run in a massively parallel fashion. In a typical application, an individual shader program will be loaded onto a GPU and many thousands of instances of the program will be executed, with varying input parameters. The returned values of each instance are then returned to the 'host-side' of the application, allowing for the optimisation of tasks that are well suited to parallelisation.

A ray generator shader spawns rays to sample the environment with – these are dependent on the type, position, and orientation of the chosen camera. In the case of a panoramic camera (in the display pipeline), for instance, rays are spawned in a spherical manner around a central point (the *position* component of the currently selected camera's ray generation configuration record). In the case of a compound eye (in the ommatidial pipeline), each ray is generated as a function of the relative position and direction of a single ommatidium to the eye's position and orientation (note that the 'eye'-type data records in *Figure 4* contain links to per-eye ommatidial arrays that define the ommatidial configuration of the eye).

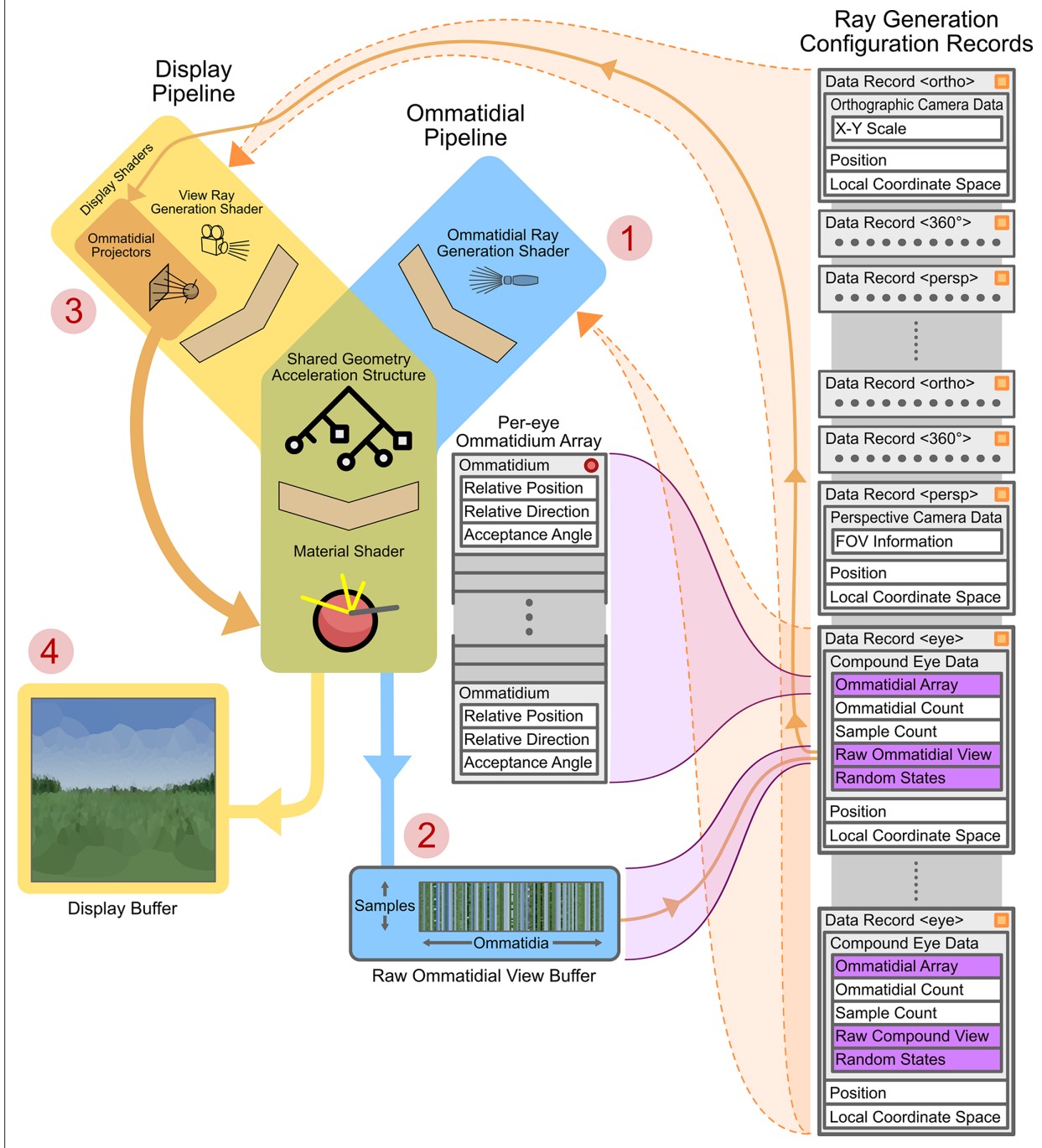

**Figure 4.** Graphics processing unit (GPU)-side renderer operational diagram. Featured are the joined *ommatidial* and *display* rendering pipelines (including display buffers) as well as the on-device (running on the GPU) per-camera configuration records and related memory allocations. Data structure memory allocations are shown in grey (indicating in-place data) or purple (indicating pointers to other sections of data). Vertical stacks of memory indicate contiguously allocated structures. Data structures marked with an orange square are automatically updated as their host-side (running on the CPU) copies change; those with red circles are manually updated via application programming interface (API) calls. Circled numbers (**1–4**) indicate the path of processing through both pipelines required to display a compound eye view. FOV: field of views.

A GAS stores the 3D geometry of the scene on the device in a format optimised for ray-geometry intersection and also provides functions to perform these intersection tests. Upon the intersection of a generated ray and the environment the material shader associated with the intersected geometry is run. These material shaders are responsible for the calculation and retrieval of the correct colour for the given geometry. For example, if a ray were to fall onto a blade of grass, the material shader

would be expected to compute and return the colour green. We note that CompoundRay currently implements a simple texture lookup, interpolating the nearest pixels of the texture associated with the geometry intersected. Future extensions are presented in more detail in the 'Discussion section.

Two pipelines are used in the renderer: an ommatidial pipeline and a display pipeline. The ommatidial pipeline handles sampling of the environment through the currently active compound eye, saving all samples from all ommatidia to a buffer stored within the eye's 'raw ommatidial view' portion of its data record. First, it generates per-ommatidium rays (*Figure 4* -1) via instances of the ommatidial ray generation shader. It then uses these to sample the environment through the GAS and appropriate material shaders, finally storing each ommatidium's view as a vector of single-pixel tri-colour samples in the raw ommatidial view buffer (*Figure 4* -2).

Conversely, the display pipeline handles the generation of the user-facing display buffer (*Figure 4* –4). In the case of generating a compound eye view display, *ommatidial projector* shaders are spawned (one per output pixel) and used to lookup (orange arrow connecting *Figure 4* parts 2 and 3) and average the values associated with each given ommatidium from the per-camera raw ommatidial view buffer. These values are then re-projected into a human-interpretable view – in the case of the image seen at *Figure 4* –4, using an equirectangular orientation-wise projection of the Voronoi regions of each ommatidium. However, for practical applications, this re-projection can be to a simple vector of all ommatidial values, allowing an algorithm to examine the data on a per-ommatidium basis, much in the same way that a real neural circuit would have direct access to the light sensors within the eye. In the case where the current camera is non-compound the display pipeline becomes the only active pipeline, handling the initiation of ray generation, GAS intersection, and materials shading for any given camera view in a single pass using a standard 2D camera projection model (simple pinhole, equirectangular panoramic, or orthographic). By referencing both with GPU video and RAM pointers, both pipelines are able to share a common GAS and material shaders so as to save device memory.

A full compound eye rendering (as indicated via numbered red circles in *Figure 4*) consists of first rendering the ommatidial view via the ommatidial pipeline to the current camera's raw ommatidial view buffer using the associated eye data record. After this, the raw ommatidial view buffer is re-projected onto the display buffer as a human-interpretable view via the ommatidial projector shaders within the display pipeline. Alternatively, for debugging purposes, rendering can be performed using a traditional camera rather than via an eye model, skipping steps 1 and 2 and instead simply rendering to the display view by passing fully through the display pipeline using the data record associated with the selected camera.

## Scene composure and use

Using the glTF file format for scene storage allows the experimental configuration of 3D models, surface textures, and camera poses to be packed into a single file (although per-eye ommatidial configurations are stored separately as comma-separated value (CSV) files and linked to within the glTF file to avoid file bloat, retaining the readability of the glTF file). The glTF format is a JavaScript object notation (JSON)-based human-readable file format for 3D scene storage that is readily supported by a number of popular 3D editors and has allowances for extensions within its specification. As such, all configurable parts of the scene that relate to CompouundRay are placed within each camera's reserved 'extras' property. The renderer is programmed to ingest these extra properties, expanding the scene definition beyond standard glTF without compromising the readability of scene files by other third-party 3D modelling suites.

While the software comes packaged with a stand-alone executable that can load and render compound eye scenes, this is not the most powerful way of using the tool. When compiled, a shared object library is also generated that can then be called from more accessible languages, such as Python (*Van Rossum and Drake, 2009*). Through the use of this, Python's *ctypes* system and a helper library that is bundled with the code, can be used directly with Python and the Numpy (*Harris et al., 2020*) mathematical framework. This allows for a significantly wider experimental scope through automated configuration – for instance, one can programmatically move an insect eye around a simulated world extracting views directly from the renderer to be used alongside neural modelling frameworks accessible under Python. This is the recommended way of using the rendering system due to the increased utility that it affords.

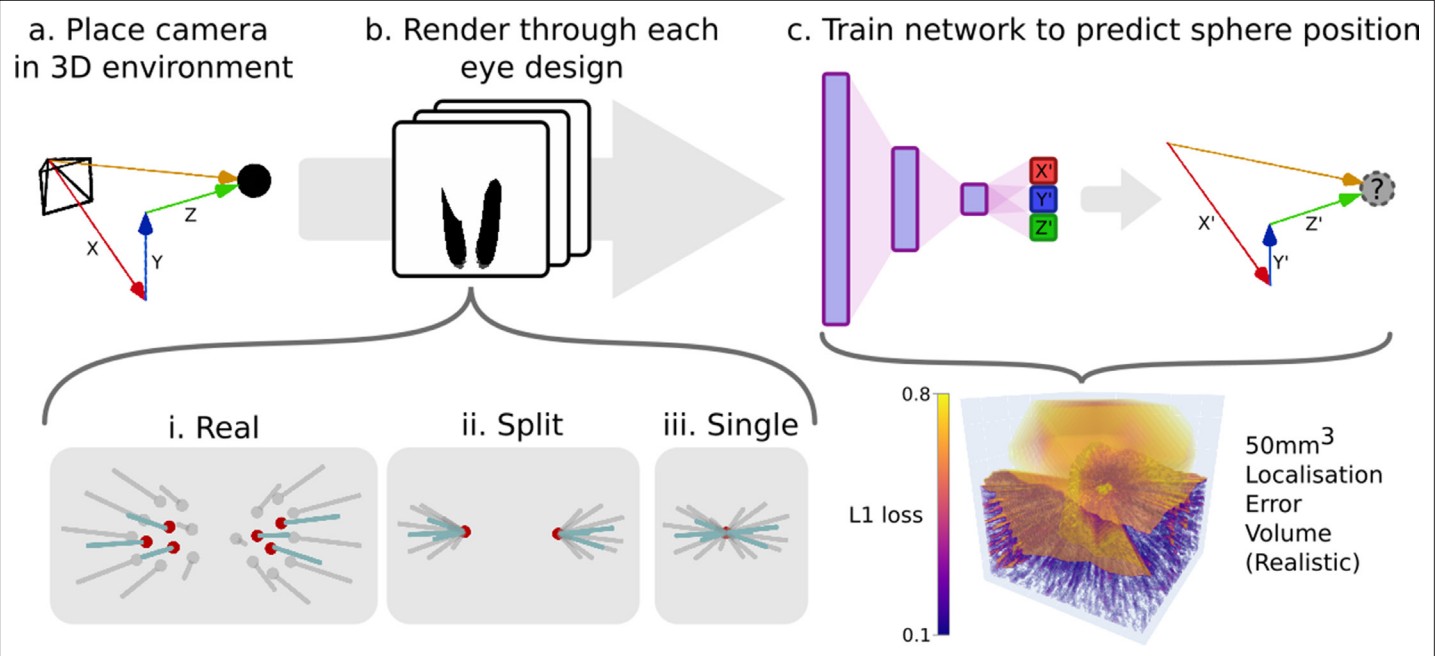

**Figure 5.** An example pipeline for exploring eye designs leveraging CompoundRay's ability to produce large quantities of data. (**a**) A camera is placed at various locations within an environment consisting of a single small point in an entirely white world. (**b**) Images (and the associated relative position of the camera in the $x$, $y$, and $z$ axes) are collected from each location through the differing eye designs that are being compared (here 'realistic', 'split', and 'single' designs). (**c**) A simple multi-layer perceptron neural network is trained to predict the the relative position ($x'$, $y'$, and $z'$) given a compound eye view. The quality of this encoding can then be interrogated by sampling uniformly across the test volume forming an error volume, here presented directly as the L1 loss (the sum of all absolute errors between the output of the network and each axis of the ground truth relative position vector) for any given point.

## Example inter-eye comparison method

To demonstrate the utility of CompoundRay for modern, data-intensive scientific analysis, we designed an example experiment. We note that any data outcomes are secondary to the validation of the tool and experimental procedure. Three virtual eye models were created, based on increasing simplifications of real-world eye data. The virtual eye models were assembled based on one (*Apis mellifera*) of eight scans of real-world samples (two *Apis mellifera*, six *Bombus terrestris*) taken by *Baird and Taylor, 2017*, with the 3D surface shape used to derive sample points and axial directions for each of the 6374 ommatidia present in both eyes. This model was used as a baseline 'real' model, from which two simplified derivatives, 'split' and 'single' were created. These were created by editing the eye's ommatidial positions to either converge on two or one single point (*Figure 5 i-iii*, i-iii) to mimic assumptions made in past simulated compound vision studies (*Franz et al., 1998*; *Baddeley et al., 2012*; *Nityananda et al., 2016*; *Polster et al., 2018*). The three variations on the real eye design were then assessed against each other for their ability to resolve the 3D position of a target point within a simple black and white environment by training a multi-layer perceptron (MLP) network with two fully connected hidden layers to estimate the point's position, encoding the relationship between visual scene and relative location - a task that was taken as a proxy for the eye's ability to perform basic small-point spatial awareness tasks. The accuracy of the network's ability to represent this encoding at any given point within a sampling volume (forming an error volume as seen in *Figure 5c*) could then be compared to another, producing maps of difference in utility of a given eye design over the sampling volume when compared to another (Figure 11b, c). The results of this experiment are presented in the later section 'Example experiment: *Apis mellifera* visual field comparison'.

Comparing the views from varying ommatidial eye designs presents a number of challenges. In particular, the differing surface shapes and ommatidial counts between any two eyes make direct comparison very challenging as a result of the lack of a one-to-one correspondence between the ommatidia of each eye. Due to these challenges, the method of indirect comparison via a proxy variable was chosen. In this example comparison method, we propose using the task of small-point localisation

as a basis for measuring eye utility, as similar tasks have formed key parts of many behavioural studies (*van Praagh et al., 1980*; *Juusola and French, 1997*; *Nityananda et al., 2016*). As shown in *Figure 5*, a camera simulating the desired eye design is placed within a matte white environment with a small matte black sphere (in this case, 2 mm in diameter) placed at its centre. The camera can then be moved around a sampling volume (here 50 mm$^3$), and a pair consisting of the view from the eye and the relative $x$, $y$, and $z$ offset, forming its *relative position*. A neural network can then be trained on a sufficient number of pairs encode the mapping from eye view to relative position (localisation).

The non-Euclidian surface geometry of the compound eye makes convolutional neural network architectures (which typically operate over euclidian 2D or 3D space [*Li et al., 2021*]) inappropriate, as doing so would require a conversion from eye surface to a 2D surface, potentially introducing biases (as further elaborated in the 'Discussion' section). Graph neural networks (*Scarselli et al., 2009*) – neural networks that operate independent of assumptions of implicit spatially uniform sampling, instead operating over data structured in a graph – provide a method of operating over the 3D surface of the eye. However, the simpler approach of flattening the visual field into a single vector (with each element representing a single ommatidium's view sample) was taken here. This approach chosen as the task of self-localisation from a single dark point in a light environment becomes a simple pattern matching task due to every position in the field producing a unique view, not requiring translation or rotation invariant feature recognition, all of which can be encompassed in a simple fully connected MLP network with two or more hidden layers (*Lippmann, 1987*). Furthermore, the views were passed to the network as one-dimensional black and white images to reduce on computational cost when running the experiment, as the environment was purely black and white, resulting in the red, green, and blue components to each ommatidial sample being identical.

For the purposes of our example experiment any encoding sufficiently capable of representing the mapping of visual input to relative location could have been used. The example MLP network we provide is specific to the example scenario given and as such should only be referred to as a starting point when considering a neural-network-based abstracted comparison method. We designed an MLP with 2 hidden layers (1000 and 256 neurons, with rectified linear activation functions to act as non-linearities) and a final linear output layer of 3 neurons to represent the x, y, and z components of the relative position vector. The hidden layers act to encode the non-linear relation between the views and the relative position of the eye, with the count at each layer chosen to provide a gradual decline in complexity from input visual field size to the 3D output vector.

The data was standardised using z-score normalisation across all training points for each eye on both input and output dimensions before training to ensure learning was weighted equally over all dimensions of the input vector and to reduce bias on output vectors. As each eye sample had a differing number of ommatidia, the size of the input vector (and as such the first layer weight vector) for the neural network differed from eye to eye. Batched stochastic gradient descent was then used to optimise the sum of all absolute errors between the output of the network and each axis of the ground truth relative position vector (the *L1 loss function*). This error could then be observed with respect to the experimental sampling volume to assess the eye design's task-specific performance across the sampling space (*Figure 5c*). A batch size of 64 was selected in order to train at an appropriate speed while balancing efficient use of available hardware (*Golmant et al., 2018*); however, in practice, any commonly used batch size would be suitable.

For each eye design, 100,000 image position pairs were generated for training purposes. Of these, 80,000 were used for training, and 20,000 were used as a validation dataset in order to plot training progress. For each derivative eye design, the neural network was trained across 100 epochs (Figure 11a shows the error over the validation set during training). A total of 100 epochs were chosen as by this point in training no significant further training gains are made, as can be seen inFigure 11a.

## Results

The analysis that follows first assesses the performance of CompoundRay with respect to the three criteria defined in the 'Introduction', before showing its utility in an example experiment. Performance is benchmarked in two 3D environments: a lab environment inspired by those used in insect cognitive experiments (*Ofstad et al., 2011*); and a 3D scan of a real-world natural environment covering an area of a number of square kilometers. The natural environment was constructed using structure-from-motion photogrammetry using photos captured from a drone over rural Harpenden, UK (3D model

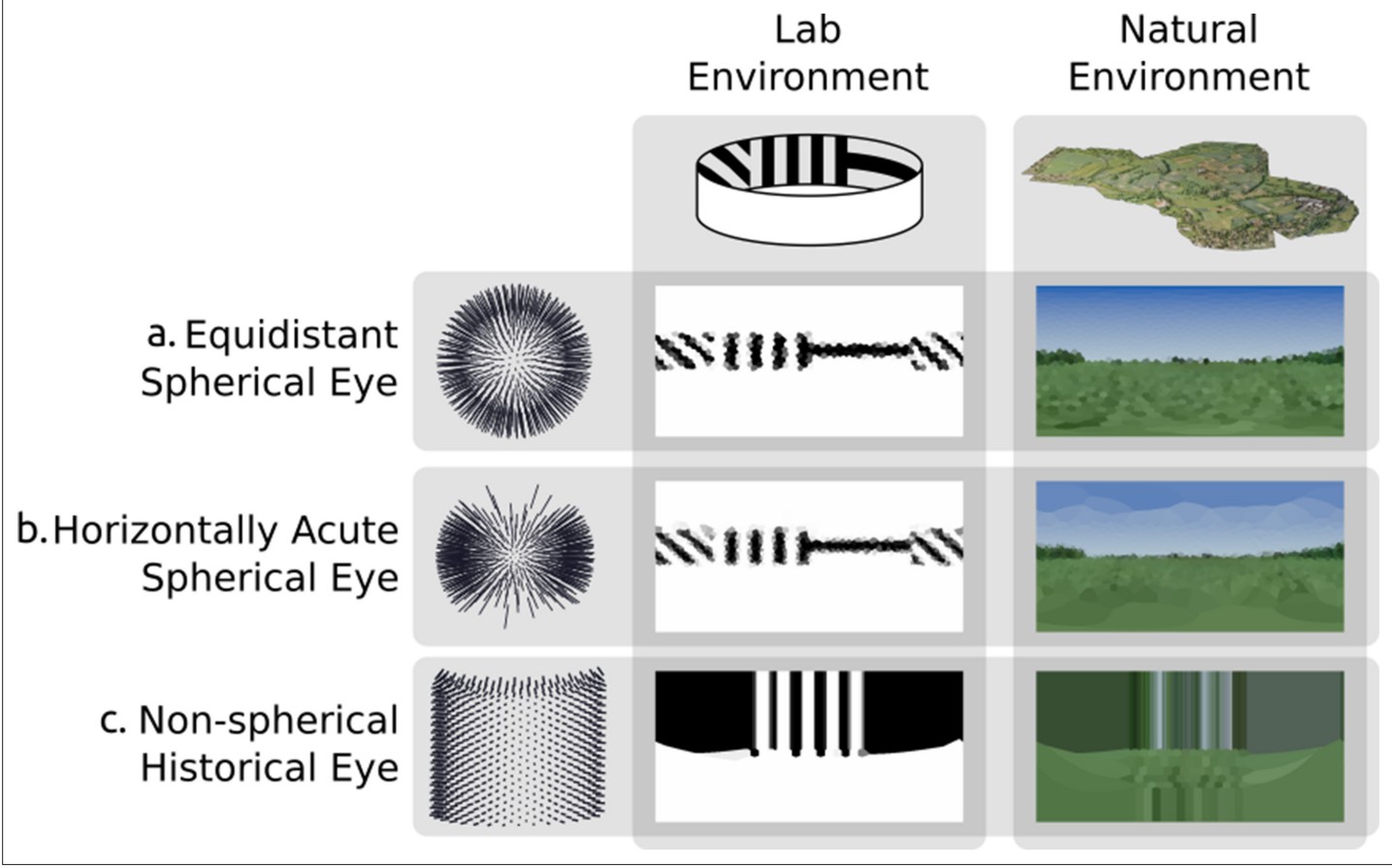

**Figure 6.** Rendering from eyes with ommatidia placed on arbitrary surfaces in one lab environment and one three-dimensional scan of a natural outdoor environment. Renderings are projected to two-dimensional using a spherical orientation-wise Voronoi latitude/longitude mapping (*orientation-wise equirectangular spherical mapping*). (**a**) Spherically uniformly distributed ommatidia. (**b**) Ommatidia clumped to form a horizontal acute zone (**c**) A design inspired by the *Erbenochile erbeni*.

subject to upcoming publication, available via contact of Dr. Joe Woodgate, Queen Mary University of London).

## Criterion 1: arbitrary arrangements of ommatidia

Criterion 1 states that the renderer must support arbitrary arrangements of ommatidia within 3D space. *Figure 6* shows two different environments rendered using three differing eye designs, from a simple spherical model with spherical uniformly distributed ommatidia that more closely aligns with current compound eye rendering methods, to a hypothetical arbitrary arrangement similar to that of the now long-extinct *Erbenochile erbeni*. While *Figure 6a&b* is still possible to simulate using traditional projection-based methods due to their ommatidial axes converging onto one central spot, 6 c demonstrates an eye with multiple unique focal points across an irregular surface which must be rendered from the viewpoint of each ommatidium independently.

## Criterion 2: inhomogeneous ommatidial properties

Criterion 2 states that it should be possible to specify heterogeneous ommatidial optical properties across an eye, such as the enlarged facets as found in robberflies (*Wardill et al., 2017*). As demonstrated in *Figure 7*, the renderer achieves this by allowing for shaping of the acceptance cone of each ommatidium. In doing so, *Figure 7* also demonstrates the importance of heterogeneous ommatidial configurations.

*Figure 7a&b* shows a non-uniform eye design with homogeneous, globally identical, ommatidial acceptance angles. It can be seen that in *Figure 7b*, where the acceptance angle is lower, aliasing occurs

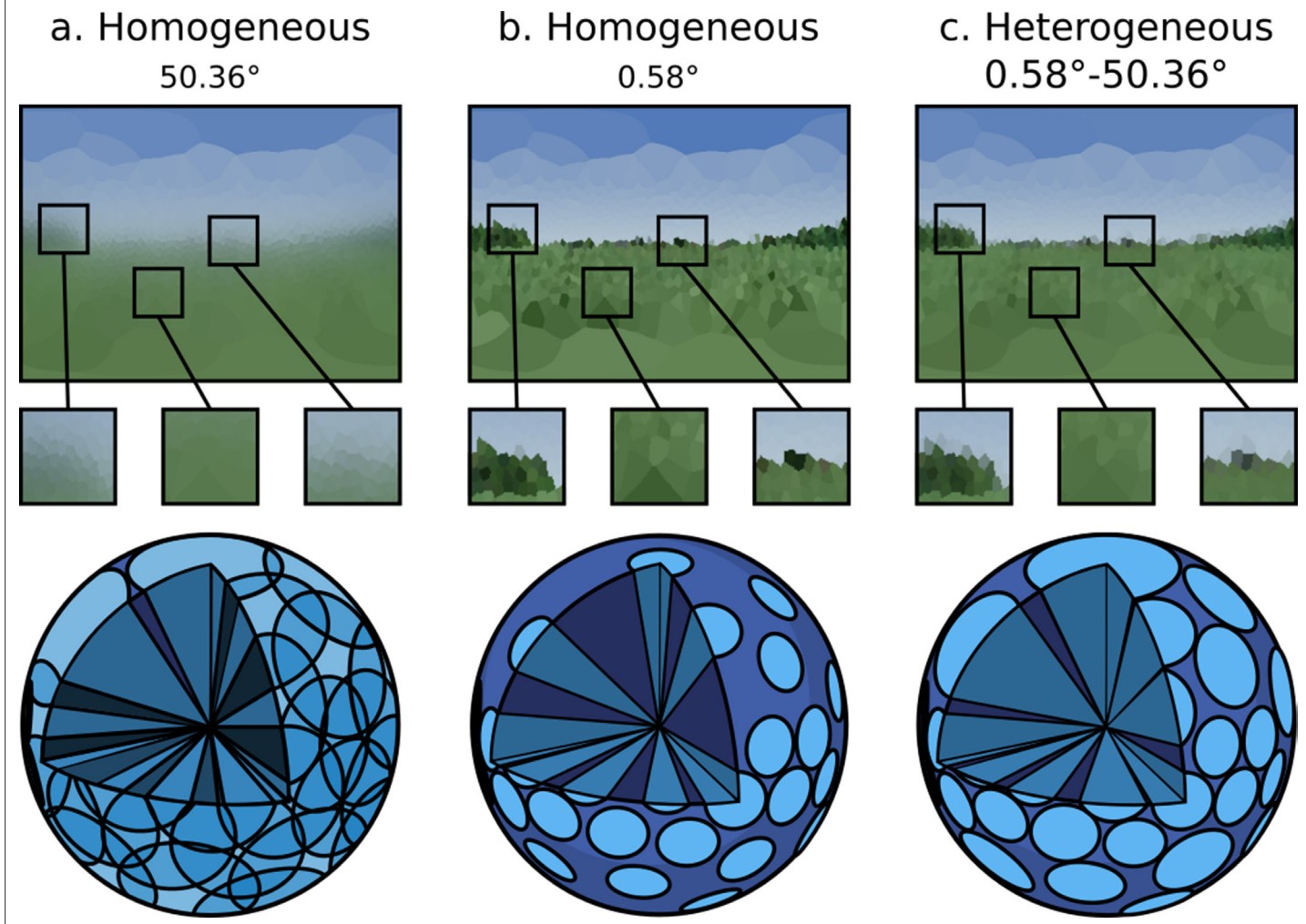

**Figure 7.** A single environment as viewed through: (**a** and **b**) eye designs with homogeneous ommatidial acceptance angles, note the blurred edges in (**a**) and sharp aliased edges in (**b**). (**c**) An eye with heterogeneous ommatidial acceptance angles. Below each output image is a three-dimensional depiction of the field of view of the specific array of ommitidia making up each eye: (**a**) oversampled, homogeneous eye; (**b**) undersampled, homogeneous eye; (**c**) evenly sampled, heterogeneous eye. Model (**c**) has the benefit of being able to leverage the most useful aspects of (b) and (a). Renderings are projected to wo-dimensional using an orientation-wise equirectangular spherical mapping.

as objects are only partially observed through any given ommatidium, causing blind spots to form, introducing noise into the image. Conversely, in *Figure 7a*, where the acceptance angle is large, the image becomes blurry, with each ommatidium oversampling portions of its neighbour's cone of vision. *Figure 7c*, however, demonstrates a heterogeneous ommatidial distribution, in which the acceptance angles toward the dorsal and ventral regions of the eye are larger compared with the angles of those around the eye's horizontal acute zone. This mirrors what is seen in nature, where ommatidial FOV is seen to vary in order to encompass the entirety of an insect's spherical visual field (*Land and Nilsson, 2002*). As can be seen in the generated image, this is important as it appropriately avoids creating blind-spots in the dorsal and ventral regions while also avoiding oversampling (blurring) in the horizontally acute region, resulting in a much clearer picture of the observed environment (before the visual data is passed into any insect visual processing neural circuits) while also minimising the required number of ommatidia. The data presented in *Figures 6 and 7* clearly demonstrates differences in visual information provided by different ommatidial layouts, reinforcing the need for realistic eye modelling methods.

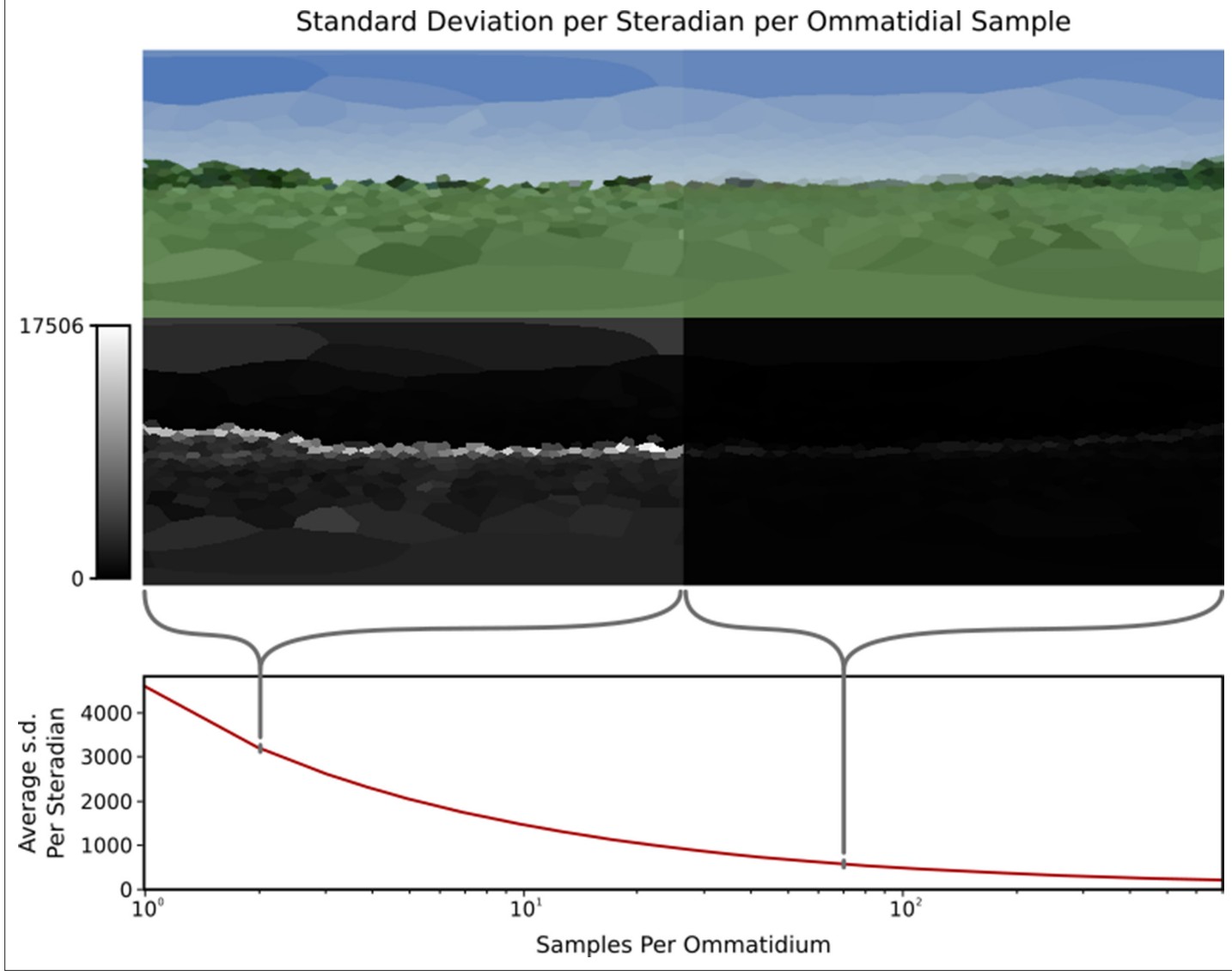

**Figure 8.** A single viewpoint viewed using variable samples per ommatidium. Top: How changing sample count changes the generated image. Note that the left side (with only 1 sample per ommatidium) is more 'jagged' than the right side image (70 samples per ommatidium) due to aliasing. Middle: Relative standard deviations of each ommatidium across 1000 frames, where difference is measured as the Euclidean distance between the two colours, this highlights the significantly larger standard deviation at points of high visual contrast in the image. Bottom: Plot of the average standard deviation of an eye per sample rays per ommatidium, normalised to one steradian. The standard deviation decreases as per-ommatidium sampling rate increases.

## Criterion 3: speed

### Minimum sample count

Criterion 3 states that to rapidly explore the information content of various eye designs any such rendering system should be able to perform at real time or faster. However, speed of the system is dependent on the number of rays required to render a given scene from a given eye; higher total ray counts will require more compute power. As discussed in the section 'Modelling individual ommatidia', per-ommatidial sampling choice can have a dramatic impact on the image generated, meaning that lower ray counts – while increasing speed – will decrease the accuracy of the image produced. As a result, it is important to be able to define some minimum number of samples required for any given eye in any given environment in a reliable and repeatable manner. Previous works *Polster et al., 2018* have attempted to establish a baseline number of samples required per ommatidium before additional samples become redundant only via qualitative visual analysis of the results produced. Here we

perform quantitative analysis of images produced by the renderer in order to more definitively define this baseline (*Figure 8*).

Unlike previous works, the temporally stochastic nature of *CompoundRays*' sampling approach can be used to aid the measurement of impact that sampling rate has on the final image. By measuring the per-ommatidium spread (here, standard deviation) of the Euclidean distance of the received colour as plotted in RGB colour space over a range of frames captured from a static compound camera, the precision of the light sampling function approximation can be measured. To control for the varying FOV of each ommatidium (which impacts the number of samples required, as larger FOVs require more sampling rays to accurately capture the ommatidium's sampling domain), we normalize the measure of spread recorded at each ommatidium against its degree of coverage, in steradians, expressing the solid angle subtended by the ommatidium's FOV at the eye's centre. That is, a steradian measures the solid angle within a sphere, in the same same way that a radian measures 2D angles within a circle.

*Figure 8* shows this measure of variance within the compound eye across a range of sample-rays-per-ommatidium, from 1 to 700. As the number of samples increases, the variance decreases logarithmically. We propose iteratively increasing samples until the maximum variance falls below a defined threshold value. Here, we use a threshold of 1%, meaning that the maximum expected standard deviation of the most deviant ommatidium on an eye (normalised to one steradian) should be no greater than 1% of the maximum difference between two received colours, here defined as the length of vector [255, 255, 255]. This method allows for standardised, eye-independent configuration of per-ommatidial sampling rate.

However, the required number of samples per eye is highly positively correlated with the visual frequency and intensity of the simulated environment at any given point, with locations exhibiting

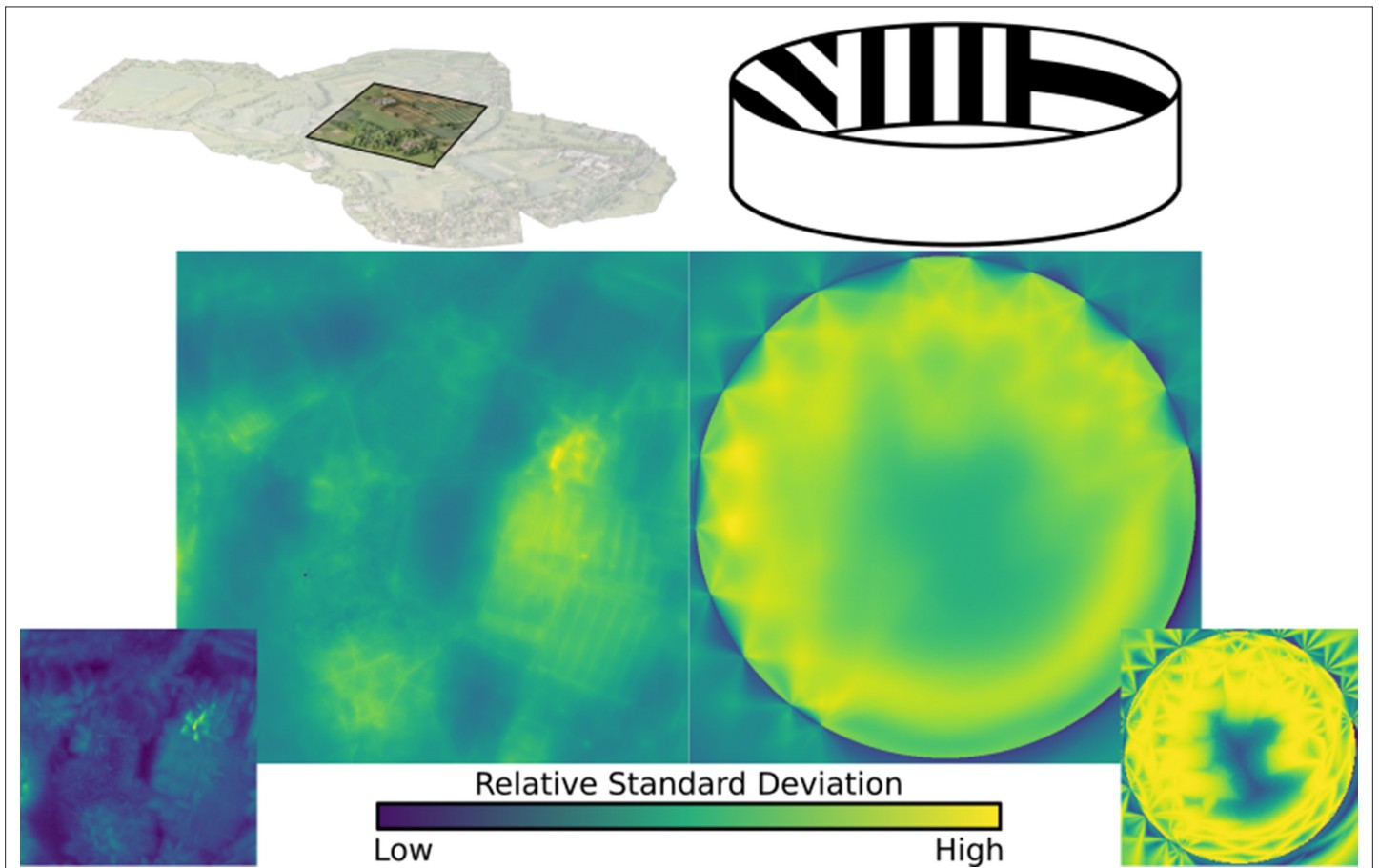

**Figure 9.** Standard deviation, averaged over all ommatidia in an eye, mapped over the natural and lab environments, showing a range of 'hotspots' of high variance in renderings. These hotspots must be accounted for when calculating minimum sample ray count. Inserts bottom left and right: the maximum (over all ommatidia) standard deviation recorded in the eye at each location.

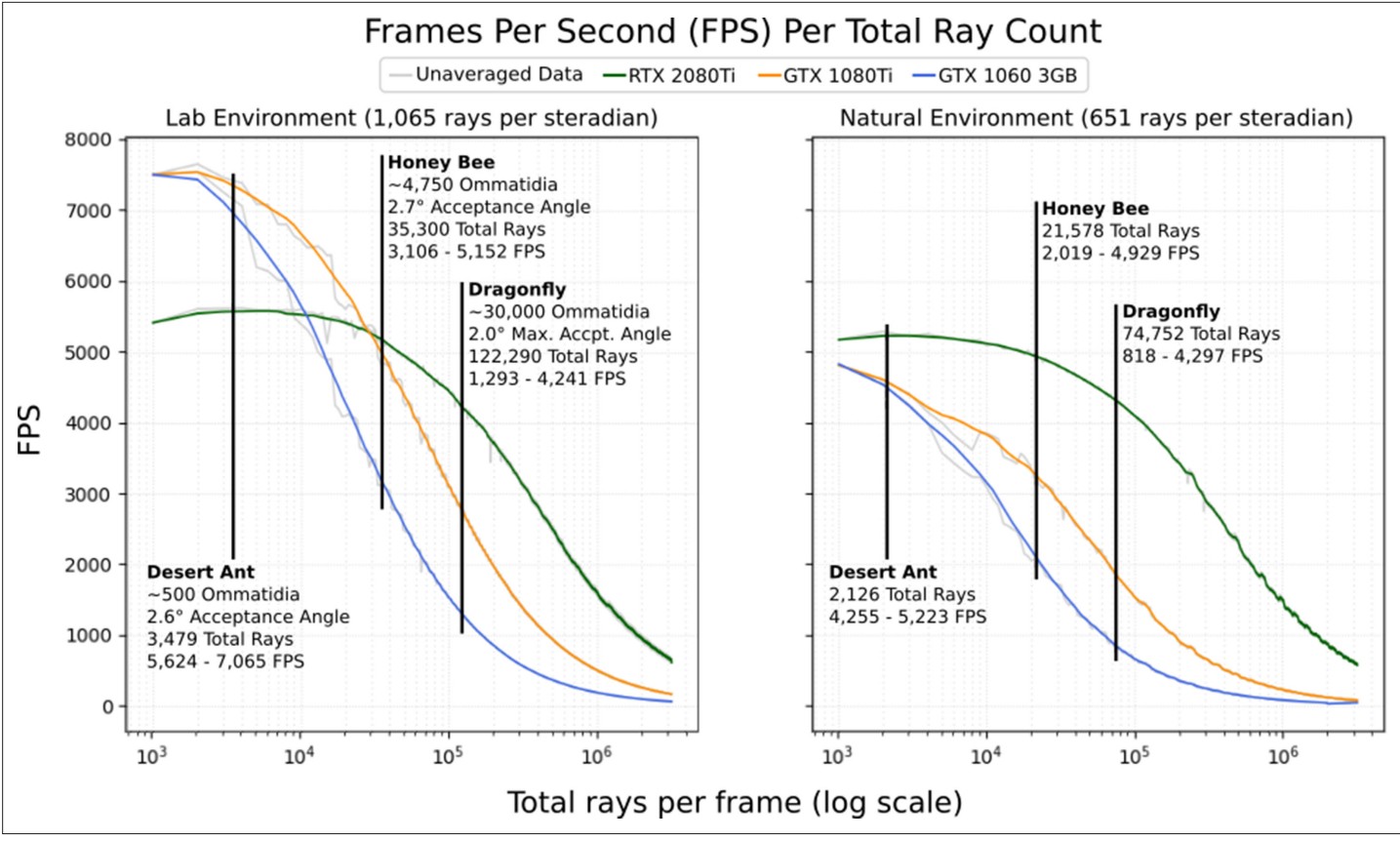

**Figure 10.** The average frames per second (FPS; 500 samples) per total number of rays dispatched into the scene for 3 different graphics cards in the 2 environments depicted in *Figure 6*. Marked are expected FPS values for a desert ant (*Schwarz et al., 2011*), honey bee (*Greiner et al., 2004*), and dragonfly (*Labhart and Nilsson, 1995*; *Land and Nilsson, 2002*). Averaged data is smoothed using a 1-dimensional box filter acting over 10 samples, clipped at the beginning and end.

high contrast seen from a distance requiring higher sampling rates in order to properly resolve. This can be seen in the skyline in *Figure 8*, which remains dominant throughout the majority of the sampling rates, consistently forming the source of the highest spread. *Figure 9* shows how the perceived variance changes over two example environments. The varying environment presents an obstacle in our attempts to standardise sampling rates, as deriving a suitable minimum sampling rate at one point in an environment – e.g., where the visual frequency was, on average, low – could lead to aliasing occurring in more visually complex areas of the same environment, potentially introducing bias for or against certain areas if analysis on the generated data were sensitive to the variance introduced.

Ultimately, the varying visual complexity poses a challenge. In an ideal scenario maximum per-ommatidium frame-to-frame variation caused by visual complexity should be a controllable factor. In the examples presented in this paper, we worked to minimise this variation to a point at which the most variable ommatidial response had a standard deviation of 1% or less of the total maximal difference in colour. One way of achieving this would be to keep a running measure of the standard deviation of each ommatidium and dynamically adapt sampling rates to ensure that no ommatidium had such a high sampling spread (adaptive antialiasing). In this work, however, we define a minimum sampling count for a given environment by performing a search for that environment's point of highest visual complexity (indicated by highest per-ommatidium deviation) – with this location found, sampling rate can be increased until deviation decreases to acceptable limits (in this case, within 1%). *Figure 9* shows plots of the relative variation in standard deviation over the lab and (partial) natural environments.

## Performance

*Figure 10* shows the rendering speed, in frames per second, against the number of sample rays being emitted in each frame when running using a selection of high-end consumer-grade graphics cards and our two sample environments. Marked on the graph are the total per-frame sample counts required by common insect eyes, chosen to encourage a maximum per-ommatidium standard deviation of 1%.

As can be seen, performance improves significantly when the renderer is able to utilise the next-generation RT hardware available in the newer RTX series of NVidia graphics cards, consistently outperforming older cards in the larger natural environment and outperforming in the smaller lab environment when using more than 30,000 rays. Counterintuitively, the previous generation of NVidia graphics cards was found to outperform the RTX series in the (relatively small) lab environment at lower ray counts (below 30,000). This appears to be due to static overhead involved with the use of the RTX series' RT cores, which may also be the cause of the small positive trend seen between 1000 and 8000 samples when using the 2080Ti. Another potentially counterintuitive result that can be seen here is the higher performance of the dragonfly eye in the natural (high polygon) environment versus in the low-polygon laboratory environment. This is because the laboratory environment produces significantly higher visual contrast (emphasised in *Figure 9*) due to its stark black/white colour scheme, resulting in a requirement of almost double the number of rays per steradian to target 1% standard deviation, resulting in higher ray counts overall, particularly in models with a high number of ommatidia.

## Example experiment: *Apis mellifera* visual field comparison

Leveraging the speed of CompoundRay, an example data-driven comparative analysis was performed between a realistic eye model and two simplified derivatives. We note that the data outcomes should be considered as pilot data, with the focus being on proving the tool and its usability. As laid out in the section 'Example inter-eye comparison method', this experiment used an MLP network to estimate the relative position of a 2 mm sphere within a 50 mm$^3$ test area using only the visual information provided from each of the eye models, with neural network performance acting as a measure of eye configuration utility. The models were split into the 'real' eye model (*Figure 5i*), an eye pair based directly on the measurements of *Baird and Taylor, 2017*; the 'split' eye model (*Figure 5ii*), an eye pair retaining per-ommatidial headings (ommatidial axial directions), but sampled from the centre points of each eye (as per approaches common in insect binocular vision study); and finally the 'single' eye model, an eye pair sampled from a single point between the two eyes (*Figure 5iii*). We hypothesise that the 'real' eye design will outperform the 'split' and 'single' eye models by increasing margins, as for each of the latter two designs the data relating to the surface shape and relative positions of the eyes is increasingly reduced.

In terms of total error over the validation set, we observed that – as expected – the 'real' eye design appears to offer a higher utility (lower total error when trying to locate the relative position of the sphere), ultimately decreasing its network's validation error quicker and to a lower minimum after 100 epochs when compared to both the 'split' and 'single' eye designs (*Figure 11a*). Interestingly, despite an initial slow in learning rate, the 'single' eye design eventually achieved a similar utility score on the validation set than that of the 'split' eye design.

To qualitatively assess the performance of each eye across the entire 50 mm$^3$ sampling cube, a further 1,000,000 samples were taken at regular intervals forming an error volume for each eye design by measuring the error between the trained neural network's relative position and the real relative position (the error volume for the 'real' eye is pictured in *Figure 5c*). In this case, to demonstrate versatility, CompoundRay was used in real time to generate these views, rather than simply being used in an offline manner to generate these benchmarking datasets.

The most obvious region of high error is that outside of the insect's FOV. As these eyes were designed with uniform average acceptance angles, this is larger than might be expected of a real bee eye. Comparing the 'real' eye to its 'single' and 'split' counterparts by means of comparing the absolute difference between their error volumes (creating the difference volumes shown from top-down in *Figure 11b&c*), patterns emerge. Seemingly, the most significant region of difference between the 'real' eye model and the 'single' eye model can be seen around the edges of the eye's field of vision. As this difference follows the entirety of the edge of the eye's field of vision, this appears to be likely due to parallax – the 'real' eye samples from two points, allowing it slightly

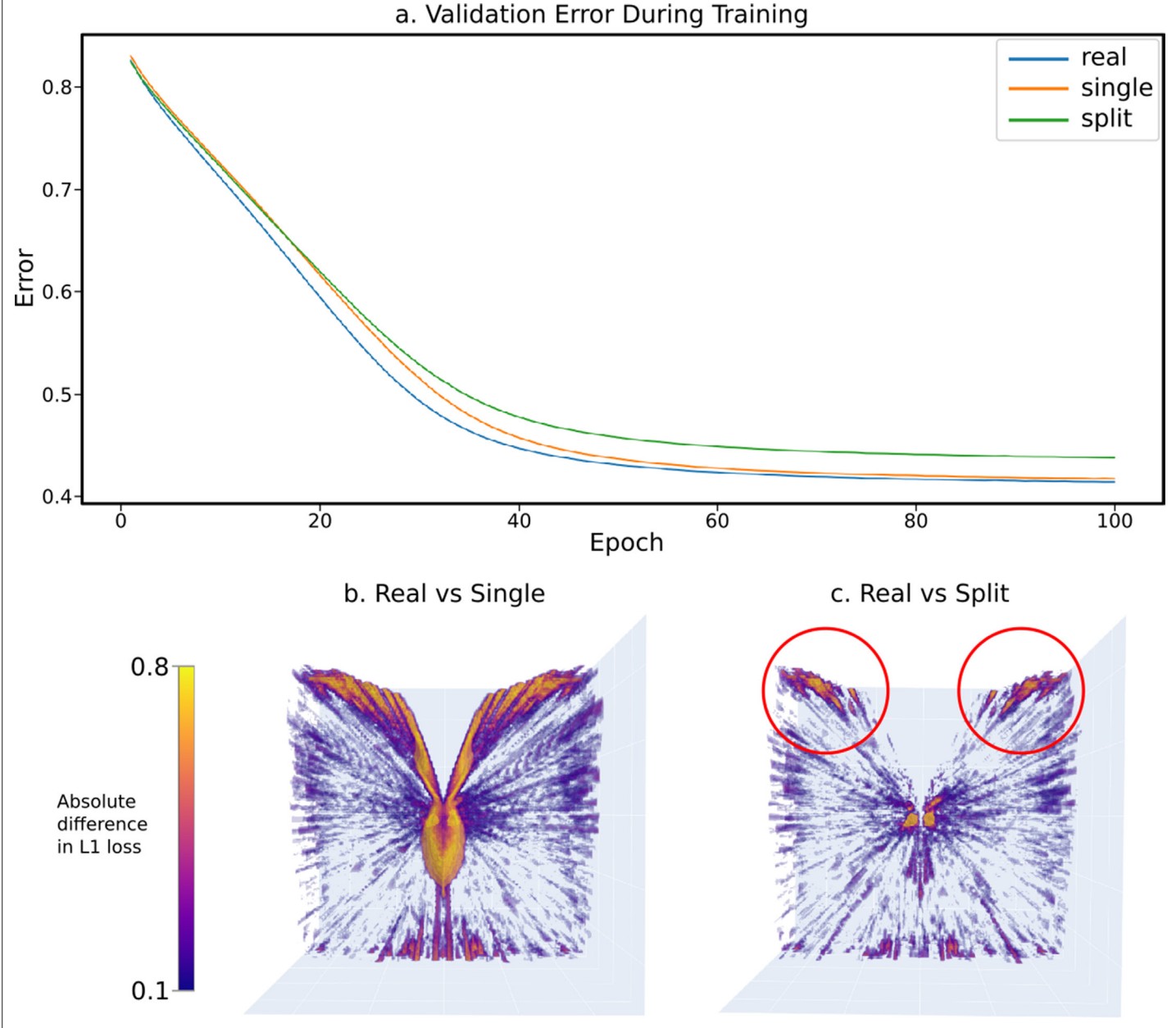

**Figure 11.** The results of a sample experiment comparing the relative effectiveness of three progressively simplified dual-eye designs – a 'real' design built from data collected from real-world *Apis mellifera* (**Baird and Taylor, 2017**) and two progressively simplified designs: the 'split' design, collapsing ommatidial viewing points onto two eye centres, and the 'single' design, projecting all ommatidial viewing points onto one central position (figurative examples shown in **Figure 5i–iii**). (**a**) The validation error graphs over 100 epochs of training for each eye design. The majority of learning occurs between epoch 0 and 60, and by epoch 100 all networks have converged in optimal-approximating states. (**b**) The absolute relative differences across the error volumes (see **Figure 5c**) between the 'real' and 'single' eye designs, as seen from the top. (**c**) As in (**b**), but between the 'real' and 'split' designs.

extended range of vision compared to that of the 'single' eye model, resulting in slightly different ranges of FOV between the two, with this difference increasing the further from the eye the point is. In the 'real' vs 'split' comparison image, however, there are two regions of higher error (highlighted in red circles in **Figure 11c**) toward the forward-facing regions of the eye's vision. These indicate a region of improved visual information when using a full eye surface as compared to simple two-point sampling of the environment.

## Discussion

This paper has introduced a new compound eye perspective rendering software, *CompoundRay*, that addresses the need for a fast, compound eye perspective rendering pipeline (*Stürzl et al., 2015*; *Taylor and Baird, 2017*; *Millward et al., 2020*). In particular, the tool supports arbitrary instantiation of heterogeneous ommatidia at any place in space and can perform rendering tasks at speeds in the order of thousands of frames per second using consumer-grade graphics hardware. We have demonstrated the utility afforded by the software and highlighted the importance of using higher-fidelity compound eye perspective rendering systems by demonstrating the visual differences introduced by altering these traits. It has also set a grounding for reasonable use of the software to ensure reproducibility and reduce biasing introduced as a result of the variety present in simulated environments and compound eye designs while warning of potential dangers that might arise during data analysis. With the introduction of the CompoundRay rendering library, it is possible to simulate in a timely manner the visual experience of structurally diverse compound eyes in complex environments, digitally captured or otherwise manually constructed. CompoundRay is provided as an open-sourced library (https://github.com/BrainsOnBoard/compound-ray).

This work lays the technical foundations for future research into elements of compound eye design that have previously been given little consideration. In particular, CompoundRay can act as the previously missing step to integrate recently catalogued real eye designs (*Baird and Taylor, 2017*; *Bagheri et al., 2020*) into mapped insect environments (*Stürzl et al., 2015*; *Risse et al., 2018*) to explore the information provided to insects in their natural surroundings. Similarly, the ability to configure ommatidial properties individually and with respect to eye surface shape enables new investigations into the benefits of asymmetric compound eyes such as those found in robber flies (*Wardill et al., 2017*) and crabs (*Zeil et al., 1986*). Furthermore, we see opportunities to model the recently reported microsaccadic sampling in fruit flies (*Juusola et al., 2017*), as well as aid with similar vision-centric neurological studies (*Viollet, 2014*; *Wystrach et al., 2016*; *Kemppainen et al., 2021*).

Our example study demonstrated the utility of CompoundRay for this strand of research. Beyond its ability to replicate insect vision in unprecedented detail, its image generation speeds facilitate use of contemporary data-driven methods to explore the eye design space. For example, we used over 3.3 million training images, which CompoundRay rendered in only 2 hr. The insights gained from more thorough analysis of the insect visual perspective—and its design relative to visual feature extraction—will help guide the development of artificial visual systems by considering not only visual post-processing steps but also the intrinsic structure and design of the image retrieval system itself. CompoundRay is explicitly designed to be highly configurable to allow members of the research community to pursue their specific research questions.

In our example study, we compared eye designs using an MLP network as a utility function, bypassing the need to directly compare the images generated by two differing eye designs. This was done as the task of directly comparing two dissimilar eye designs – either analytically using a comparison program or qualitatively via human observation – is non-trivial. Compound eyes are inherently different from our own, and the CompoundRay rendering system allows for large-scale data collection on arbitrary compound surfaces. We note that with this comes the risk of anthropomorphising the data received when directly observing it, or placing bias on one area over another when comparing programmatically. At its base level, the eye data gathered is most 'pure' when considered as a single vector of ommatidial samples, much in the same way that the neural systems of any insect will interpret the data. In contrast, all the images that we have presented here have been projected onto a 2D plane in order to be displayed. Great care must be taken when considering these projections, as they can introduce biases that may be easy to overlook. For instance, in *Figure 6a&b*, orientation-wise spherical projection mapping was used to artificially place the ommatidia across the 2D spaces, forming panoramic Voronoi plots of observed light at each ommatidium. In these projections, much as is seen in 2D projections of Earth, the points at the poles of the image are artificially inflated, giving a bias in terms of raw surface area affected toward ommatidia present in the ventral and dorsal regions of the eye. Not only do these visual distortions warp the image produced, but if that image is used as the basis for algorithmic comparison (e.g., as the input to an image difference function [*Philippides et al., 2011*]) then that too, will be effected by these projection biases. In the case where direct comparison between two eyes with entirely dissimilar surfaces is attempted, the problems can be

magnified further, as the projections of each eye will introduce biases that may interact in unexpected ways.

These projection biases highlight how fundamentally difficult it is to directly compare eye designs that do not share an underlying surface structure, unless those surfaces happen to be affine transformations of one another (they are *affinely equivalent*). For any surfaces that are affinely equivalent, the surface itself can be used as a projection surface. In this way, for instance, the outputs of two cubic eyes could be compared to each other by performing all comparison calculations on the cube's surface, interpolating between points where an ommatidia exists on one eye but not another. It is trivial to directly compare compound eye images from eyes with the same structure in terms of ommatidial placement, and eyes with surfaces that are affinely equivalent can also be compared; however, further precautions must be taken when comparing the images from two or more eyes that do not have compatible projection surfaces so as not to unduly bias a portion of one surface over any other. If direct comparison between two eye designs is required, and their surfaces are not affine transformations of each other, then intermediate surfaces to perform calculations over must be found within the interpolation from one surface to the other, with a significant amount of care taken to reduce as much as possible projection artefacts forming between the two. However, we recommend instead taking a proxy metric as demonstrated here (using a localisation task) to measure differences between eye-accessible information content.

By basing CompoundRay around the conceptually simple approach of ray-casting and adhering to open standards, the rendering system has been designed with open development in mind. In the future, the library could be extended to further enhance biological realism by adding non-visible spectral (e.g. ultraviolet) lighting (*Möller, 2002*; *Stone et al., 2006*; *Differt and Möller, 2015*) and polarisation (e.g.*Gkanias et al., 2019*) sensitivities, as well as more realistic photo-receptor response characteristics (e.g. *Song et al., 2009*). Similarly, new technologies could be rapidly evaluated by simulating the input to novel imaging systems (e.g. dynamic; *Viollet and Franceschini, 2010* or heterogeneous pixel arrays) and processing pipelines (e.g. event-based retinas; *Gallego et al., 2022*). In addition, environmental features such as shadow casting, reflection, and refractions present obvious extensions as well the introduction of features such as adaptive anti-aliasing and increasing the projections available to users. Currently, *CompoundRay* returns all output images in an eight-bit colourspace, which aligns with most contemporary display equipment. However, we note that the underlying GPU implementation operates on 32-bit floating-point variables which if exposed to the high-level Python API could improve the system's ability to represent visual sensors that experience a wider bandwidth of light (hyperspectral visual systems). It is hoped that as further efforts are conducted to map the eye structures of insects, CompoundRay will serve as a key tool in uncovering the intricacies of these eye designs in a modern, data-driven way.

## Acknowledgements

Thanks to Dr James Knight (University of Sussex) for his input and feedback on the manuscript, and Dr Joe Woodgate (Queen Mary University of London) for provision of the natural environment 3D model. Graphics cards were supplied by the Brains on Board research project (EP/P006094/1) and project partner NVidia. Grant funding was provided by EPSRC awards EP/P006094/1 and EP/S030964/1. For the purpose of open access, the author has applied a Creative Commons Attribution (CC BY) licence to any Author Accepted Manuscript version arising.

## Additional information

### Funding

| Funder | Grant reference number | Author |
| --- | --- | --- |
| Engineering and Physical Sciences Research Council | EP/P006094/1 | Blayze Millward |
| Engineering and Physical Sciences Research Council | EP/S030964/1 | Michael Mangan |

| Funder | Grant reference number | Author |
|--------|------------------------|--------|

The funders had no role in study design, data collection and interpretation, or the decision to submit the work for publication.

## Author contributions

Blayze Millward, Conceptualization, Software, Investigation, Visualization, Methodology, Writing - original draft; Steve Maddock, Supervision, Validation, Writing - review and editing; Michael Mangan, Conceptualization, Supervision, Funding acquisition, Methodology, Writing - review and editing

## Author ORCIDs

Blayze Millward (iD) http://orcid.org/0000-0001-9025-1484
Steve Maddock (iD) http://orcid.org/0000-0003-3179-0263

## Decision letter and Author response

Decision letter https://doi.org/10.7554/eLife.73893.sa1
Author response https://doi.org/10.7554/eLife.73893.sa2

# Additional files

## Supplementary files

• Transparent reporting form

## Data availability

The manuscript is a computational study, with all modelling code and data accessible on GitHub at https://github.com/ManganLab/eye-renderer. Use of the natural environment was kindly provided by Dr. JoeWoodgate, Queen Mary University of London and is subject to upcoming publication. As such, instead included in the CompoundRay repository is a stand-in natural 3D terrain model. As all models are used for demonstrative purpose, this stand-in model offers little difference to the natural model used, bar it's subjectively lower-quality aesthetics.

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
