## [Editor Report]

In this important work, the authors develop compelling new open source methods to study compound eye vision, with particular emphasis and examples in insects and appropriately supporting arbitrarily diverse spatial distributions, types and mixtures of types of ommatidia. The manuscript introduces example experiments to illustrate the use of the new methodology. This work supports future studies of invertebrate brains, a timely addition to the newly mapped connectomes of insect brains.

---

## [Decision Letter]

**Decision letter after peer review:**

Thank you for submitting your article "CompoundRay: An open-source tool for high-speed and high-fidelity rendering of compound eyes" for consideration by *eLife*. Your article has been reviewed by 3 peer reviewers, and the evaluation has been overseen by myself Albert Cardona as a Reviewing Editor and Tirin Moore as the Senior Editor. The following individuals involved in review of your submission have agreed to reveal their identity: Stephan Saalfeld (Reviewer #1); Andrew D Straw (Reviewer #2).

Essential revisions:

1) All reviewers agree that the raw ommatidial view buffer aspect of the work is the strongest piece and with the most potential to support neuroscience experiments. Expanding on this and discussing the possible applications in simulating insect vision and e.g., interpreting the neural connectomes of insect visual systems is most appropriate. Discussion of the modeling of e.g., insect compound eyes with asymmetric ommatidia is a must; one such example could be robber flies.

2) The motivation for the visualization for human observers is weak, and it is unclear of what use it can be for insect vision or vision research more generally. On surface, it seems a contradiction that the software goes to great lengths to model ommaditia with independent focal points, as necessary for the accurate modeling of each ommatidium, only to then project to a single nodal point to create a camera eye image, contrary to the main point of the work. A stronger motivation with explicit applications that aren't possible otherwise with this camera image are necessary. Otherwise, this aspect of the work ought to be downsized relative to the clearly innovative and useful raw ommaditial view buffer approach for compound eye vision research.

*Reviewer #1 (Recommendations for the authors):*

More details as they emerged, comments overlap:

98 Do you really mean that each ray has a maximum recursion depth of 0? Did you implement illumination by a light source? If rays do not recurse or trace to light sources, all surfaces are 100% diffuse and ambient and ray casting is equivalent to polygon projection with local illumination. If no illumination by an independent light source is implemented, all surfaces are 100% ambient and ray casting is equivalent to polygon projection. Please clarify.

64 – 67 The first two criteria both cover "arbitrary ommatidia geometry" which I think is one criterion.

65 Spectral sensitivity, which could be criterion 2 is not addressed as far as I understand. The renderer is purely RGB in 8-bit depth, correct? This is not discussed in section "Criterion 2" and could be a big obstacle for real applications. Is it hard to fix?

78 shadow maps are not more resource hungry than ray tracing with direct illumination, can you please explain better or remove?

84 interreflection of light can be implemented by e.g. calculating radiosity maps or doing sampling of rays cast from light sources. I do not see that CompoundRay is doing either, so it is not clear what the relationship to this work is? Can you please clarify?

Overall, I find the paragraph about ray-casting ray-tracing and the concrete relationship with CompoundRay unnecessarily confusing. As said before, I believe it would be better to explain the relevant differences of rendering methods only in how they relate to this project. I.e. how they produce an image (projection rendering by projecting polygons using painters algorithm, texture mapping and/ or colorize with local illumination model; ray-casting by identifying first intersection, texture mapping and/ or colorize with local illumination model, averaging over rays). Mention illumination, shadows (maps for each light in projection, testing for object intersection by tracing to light in ray-tracing) and global models (radiosity or photon sampling) only as possible extensions, not used in this work.

124 – 128 state the above but the language and reasoning are unclear.

227 – 229 eventually clarify that CompoundRay does not currently implement any illumination model or first order ray recursion to cast shadows, so the output would be equivalent to per ommatidium camera rendering and integration, this is way too late in the text as illumination and shading capabilities were used earlier to root for ray-casting for no practical reason.

*Reviewer #2 (Recommendations for the authors):*

Abstract:

L12-13: I think some expectation of why this might be relevant or interesting would be useful.

L19: Was there any insight that arose from doing this? Especially in regards to the idea raised above – that the shape and overall structure have been ignored?

Intro:

Figure 2: It would be helpful to explicitly indicate which approach is used in the present paper.

Figure 7, top panel and caption for top panel: If I understand it correctly, this top panel is a composite of 100s of individual images but I am confused that I do not see many, if any, vertical edges where one set of samples gives way to another. Also, the jagged effect is not particularly obvious.

*Reviewer #3 (Recommendations for the authors):*

Below I list a number of small details in need of attention.

Figure 2c could improve, at least by making it larger, or by magnifying a few adjacent ommatidia to better appreciate the effect of not sharing a focal point.

Line 124: Figure 1b is not adequate to illustrate that insect eyes "contain many hundreds or even thousands of lenses".

Line 183: "an image can be generated that captures the full compound eye view." But isn't the point that an image cannot be generated because of the lack of a shared focal point, assuming then that the brain's internal representation of the field of view will be the product of a neural network parsing the output of the many ommatidia? The "cone of influence" of each ommatidium on the final image seems … premature. Neural circuits will combine inputs from ommatidia in ways that have not been worked out in the insect literature, as far as I know. The motivation for wanting to generate an image is unclear: is this for ease of the researchers, which are human, or what is the goal?

The "raw ommatidial view buffer" seems most useful to neuroscientists. The design of a neural network that composes your "Display buffer" from the "raw ommatidial view buffer" would be a most interesting research project, particularly for the opportunity to compare it with the existing, mapped circuit wiring diagram, or connectome, of the *Drosophila* optic lobes. You could discuss this. In other words, Line 240, "These values are then re-projected into a human-interpretable view" is the potential start of a fascinating research project in its own right, if one redefines this as "fruit-fly interpretable view", as in, the brain of the fruit fly needs to make effective use of this information to successfully navigate the world.

Line 227: "Currently CompoundRay implements only a simple texture lookup, interpolating the nearest pixels of the texture associated with the geometry intersected." What would be gained by, or what is lost now relative to, implementing the full ray tracing approach?

Line 247: "Both pipelines share a common geometry acceleration structure and material shaders so as to save device memory." How is this possible? Can you hand-hold the reader a bit more to clarify how can both pipelines share the same GPU shaders? I may have missed a critical point.

Line 253: "rendering using a traditional camera skips steps 1", you mean, your software can alternatively use traditional cameras in place of modeled ommatidia? I am not sure I get it.

Line 293: "inhomogeneous ommatidial properties". An example you could cite is the eyes of the robber flies, Wardill et al., 2017 Curr Biol., where a frontal fovea with enlarged facets is reported.

Figure 6: seems to imply that overlap is not desirable? Overlap could allow for all kinds of spatial computations, from supraresolution to contrast enhancement to motion detection.

Line 309: the statements would apply only in the absence of a nervous system that would make full use of the ommatidial signals.

And, in general: aren't insect compound eyes known to present significant overlap in the field of view of each ommatidium relative to its neighbors? Figure 6c would then be the most realistic, if ignoring post-processing by the optic lobes?

Figure 7: qualitatively, the top rendered scene looks quite similar throughout. I am not able to notice much difference between the left and right ends. Figure 7 needs work to better convey your message, and to better establish or highlight the vertical relationship between the images in the top and middle and the plot at the bottom.

Line 357: "using a genetic algorithm to ﬁnd the location", where are the technical details of this approach? Parameters? Framework used, or code? Is it your own code?

Line 440: "On top of the potential biological insights CompoundRay may be able to offer", indeed, your paper would be so much stronger if you showed one such biological insight. It would then become a reference software for compound eye vision.

[Editors’ note: further revisions were suggested prior to acceptance, as described below.]

Thank you for resubmitting your work entitled "CompoundRay, an open-source tool for high-speed and high-fidelity rendering of compound eyes" for further consideration by *eLife*. Your revised article has been evaluated by Tirin Moore (Senior Editor) and a Reviewing Editor.

The manuscript has been improved but there are some remaining issues that need to be addressed, as outlined below:

Line 234: the sentence starting with "This will aid…" makes claims that, first, aren't entirely clear, and second, seem speculative and belong to the discussion.

Figure 10: the legend is far too short. Needs more detail on what each panel is, particularly for the lower panels. Also, where in the methods is the neural network described? Could be referenced here. Also, "network" is not quite the right shorthand to describe a neural network. Was it a CNN? Or generically an ANN? Or a perceptron?

Line 504: where is the Materials and methods description for the "4-layer fully-connected neural network"?

Line 532: "neural network was trained across 100 epochs three times", why? Why not 200 epochs, or 50, and 2 times, or 10? Why a batch size of 64? Also, these details are hardly worthy of opening the "Experimental results" section. Rather, they belong to the methods, alongside explanations and figures demonstrating why the set of parameters chosen are sensible and appropriate for the problem at hand.

Line 538, isn't the fact that a simple "single eye" design achieved a similar utility score concerning? Perhaps this reveals a disconnect between the test, which runs for 100 epochs (as plotted in Figure 10), and the real world, where an insect needs to take split-second decisions from the let go.

Line 565 reference to "Figure 10 bottom left" could really use the lettering of panels. In addition to a far lengthier and detailed explanation on what, exactly, is in that panel and how it was generated. And if these panels are as critical as they seem to be to evaluate the performance of the 3 eye models, then they ought to be bigger and far more richly annotated, to assist the reader in understanding what is being plotted.

Line 603: if you see opportunities for modeling the "recently reported" (2017) microsaccadic sampling in fruit flies, then why didn't you? Or is this the subject of a paper in preparation or under review elsewhere?

A question for the discussion: with its intrinsically distributed light sampling, compound eyes surely work not with frame-processing downstream circuits but rather event-based circuits, such as those of the dynamic vision sensor or silicon retina (Tobi Delbruck et al., various papers). Could you discuss the implications of asynchronous compound eye-based sensors, particularly when considering uneven ommatidial size which is not available in camera-based eyes such as in mammals or cephalopods? In a fast-moving animal such as a flying honeybee, asynchronous processing surely has advantages for motion detection and obstacle detection and avoidance, and scene recognition (i.e., flowers to land on), or better, to decode the waggle dance language of conspecifics.

Considering event-based computing could make for a much nicer discussion than the caveats of anthropomorphised, frame-based visual computing, or at least complement it well, outlining the intrinsic advantages of compound eyes.

---

## [Author Response]

Essential revisions:1) All reviewers agree that the raw ommatidial view buffer aspect of the work is the strongest piece and with the most potential to support neuroscience experiments. Expanding on this and discussing the possible applications in simulating insect vision and e.g., interpreting the neural connectomes of insect visual systems is most appropriate. Discussion of the modeling of e.g., insect compound eyes with asymmetric ommatidia is a must; one such example could be robber flies.

Thank you for this comment. We have addressed each of the suggestions as follows:

1. To more adequately demonstrate the utility of the raw ommatidial view buffer a new section, “Example Experiment: *Apis mellifera* Visual Field Comparison” has been added. This offers a more detailed insight into the process of using CompoundRay for a simple experiment analysing the information content available to specific eye designs, posed as a comparative analysis between the arbitrarily-surfaced eyes capable of being supported in CompoundRay and two common approximations of insect visual systems.

2. We have also added an example experiment that demonstrates applications,

3. Introduced a paragraph to the Discussion that outlines immediate areas in which the tool could be used (e.g. asymmetric patterns in robber flies)

2) The motivation for the visualization for human observers is weak, and it is unclear of what use it can be for insect vision or vision research more generally. On surface, it seems a contradiction that the software goes to great lengths to model ommaditia with independent focal points, as necessary for the accurate modeling of each ommatidium, only to then project to a single nodal point to create a camera eye image, contrary to the main point of the work. A stronger motivation with explicit applications that aren't possible otherwise with this camera image are necessary. Otherwise, this aspect of the work ought to be downsized relative to the clearly innovative and useful raw ommaditial view buffer approach for compound eye vision research.

Thank you for the comment. We completely agree with your statement that raw ommatidial view buffer (or, more accurately, a vectorised representation of it) is the more important data generated by CompoundRay. Our intention with the discussion of human visualisation and biassing issues was to highlight, visually, the pitfalls that can occur when attempting to compare two eyes with dissimilar projection surfaces. In addition, we hoped to demonstrate the utility of visualising the insect view in a human-readable way for debugging and sanity-checking purposes, itself with the added caveat that this is susceptible to biases arising from anthropomorphisation as well as projection, and should thus be avoided. To this end, we have made extensive edits to the Methods, Results (including an example experiment), and Discussion which we hope makes this position clearer. Specifically our discussion of visualisation, and the associated cross-comparison issues have been re-framed to focus on the impacts these may have on comparing dissimilar eye designs, making more limited use of the human visualisation scenario as an explanatory example and referencing the work performed in the newly added Example Experiment section.

Reviewer #1 (Recommendations for the authors):More details as they emerged, comments overlap:98 Do you really mean that each ray has a maximum recursion depth of 0? Did you implement illumination by a light source? If rays do not recurse or trace to light sources, all surfaces are 100% diffuse and ambient and ray casting is equivalent to polygon projection with local illumination. If no illumination by an independent light source is implemented, all surfaces are 100% ambient and ray casting is equivalent to polygon projection. Please clarify.

Yes, the reviewer is correct that CompoundRay currently uses a recursive depth of 0 (lines 176-179). As a first implementation our aim was to prove that arbitrary surface shapes could be rendered at the speed needed for modelling. More realistic light modelling could be introduced in later advances as outlined in the updated Discussion (line 545-547). Thus, there is an equivalency to polygon projection from each individual ommatidia, which is now discussed and explicitly stated in the paper (lines 136-137), along with reasoning in favour of ray-casting (lines 99-101), namely that it serves as a better platform for implementing new features (As an example, light-source simulation) should they need to be added in the future.

Also see “Changes Relating to Colour and Light Sampling” below.

64 – 67 The first two criteria both cover "arbitrary ommatidia geometry" which I think is one criterion.

These criteria are derived from previously published work, which has been re-worded to be more clearly the case.

65 Spectral sensitivity, which could be criterion 2 is not addressed as far as I understand. The renderer is purely RGB in 8-bit depth, correct? This is not discussed in section "Criterion 2" and could be a big obstacle for real applications. Is it hard to fix?

Yes, the renderer is currently RGB only (Please refer to “Changes Relating to Colour and Light Sampling”) but could be extended to include non-visible light, polarisation etc in the future (See new Discussion). 8-bit has been discussed already in response to point 3.

78 shadow maps are not more resource hungry than ray tracing with direct illumination, can you please explain better or remove?

This section has been removed as suggested.

84 interreflection of light can be implemented by e.g. calculating radiosity maps or doing sampling of rays cast from light sources. I do not see that CompoundRay is doing either, so it is not clear what the relationship to this work is? Can you please clarify?Overall, I find the paragraph about ray-casting ray-tracing and the concrete relationship with CompoundRay unnecessarily confusing. As said before, I believe it would be better to explain the relevant differences of rendering methods only in how they relate to this project. i.e. how they produce an image (projection rendering by projecting polygons using painters algorithm, texture mapping and/ or colorize with local illumination model; ray-casting by identifying first intersection, texture mapping and/ or colorize with local illumination model, averaging over rays). Mention illumination, shadows (maps for each light in projection, testing for object intersection by tracing to light in ray-tracing) and global models (radiosity or photon sampling) only as possible extensions, not used in this work.

We agree that the previous description of Raycasting was confusing and thus have streamlined the Methods section to reduce confusion on the concrete relationship between ray-casting methods and CompoundRay. We have also made a number of small changes throughout the section entitled “The CompoundRay Software Pipeline” that further clarifies the rendering process. All discussion of advanced light modelling etc has now been downsized as suggested and moved to the Discussion and introduced as future extensions.

Again, more clarification are provided in “Changes Relating to Colour and Light Sampling”

124 – 128 state the above but the language and reasoning are unclear

We have now added reasoning for the of raycasting (lines 99-101) which combined with the newly streamlined Methods, and movement of description of advanced lighting models to the Discussion address the above comment.

227 – 229 eventually clarify that CompoundRay does not currently implement any illumination model or first order ray recursion to cast shadows, so the output would be equivalent to per ommatidium camera rendering and integration, this is way too late in the text as illumination and shading capabilities were used earlier to root for ray-casting for no practical reason.

Explicit description of the equivalency to polygon projection from each individual ommatidia has been added (lines 136-137) as well as explicit statement of recursive depth (lines 176-179). Again, as the focus here is on the geometry modelling of eye models advanced lighting models are considered future work and thus introduced in the updated Discussion.

Please refer to “Changes Relating to Colour and Light Sampling” for additional information.

Reviewer #2 (Recommendations for the authors):

L12-13: I think some expectation of why this might be relevant or interesting would be useful.

Additional explanatory text added (lines 8-9)

L19: Was there any insight that arose from doing this? Especially in regards to the idea raised above – that the shape and overall structure have been ignored?

We have added an example experiment in which we compared realistic vs simple eye designs in the task of small object localisation, which covers this in depth (lines 394-467).

We have also added a sentence to the abstract noting this addition.

Figure 2: It would be helpful to explicitly indicate which approach is used in the present paper.

The approach used in this paper is now included within the figure caption.

Figure 7, top panel and caption for top panel: If I understand it correctly, this top panel is a composite of 100s of individual images but I am confused that I do not see many, if any, vertical edges where one set of samples gives way to another. Also, the jagged effect is not particularly obvious.

This image has now been updated to better communicate the differences between the two views by performing a direct comparison between two points on the sampling curve rather than showing a composite of many points – this makes the difference significantly more visible.

Reviewer #3 (Recommendations for the authors):Below I list a number of small details in need of attention.Figure 2c could improve, at least by making it larger, or by magnifying a few adjacent ommatidia to better appreciate the effect of not sharing a focal point.

A magnification inset has been added as suggested.

Line 124: Figure 1b is not adequate to illustrate that insect eyes "contain many hundreds or even thousands of lenses".

The subfigure has been changed to display only one insect eye (that of the Erbenochile), making the density of the lenses on the eye more clear, and including a zoom-section to the diagram of the ommatidium making the linkage between the diagram and the eye image explicit.

Line 183: "an image can be generated that captures the full compound eye view." But isn't the point that an image cannot be generated because of the lack of a shared focal point, assuming then that the brain's internal representation of the field of view will be the product of a neural network parsing the output of the many ommatidia? The "cone of influence" of each ommatidium on the final image seems … premature. Neural circuits will combine inputs from ommatidia in ways that have not been worked out in the insect literature, as far as I know. The motivation for wanting to generate an image is unclear: is this for ease of the researchers, which are human, or what is the goal?

Yes, you are completely correct. Our motivation is to more accurately model the light that arrives to the insect eye ready for analysis and integration into computational models, as such when discussing the “cone of influence” of each ommatidium, we are explicitly describing the light that arrives at the lens. Lines 182-188 have been updated to explicitly state the motivation for creating an image (debugging), and our newly added example experiment (lines 394-467) shows how the realistic eye outputs can be used with artificial neural networks to compare eye designs for specific tasks. This addition exemplifies our intended usage of CompoundRay.

The "raw ommatidial view buffer" seems most useful to neuroscientists. The design of a neural network that composes your "Display buffer" from the "raw ommatidial view buffer" would be a most interesting research project, particularly for the opportunity to compare it with the existing, mapped circuit wiring diagram, or connectome, of the *Drosophila* optic lobes. You could discuss this. In other words, Line 240, "These values are then re-projected into a human-interpretable view" is the potential start of a fascinating research project in its own right, if one redefines this as "fruit-fly interpretable view", as in, the brain of the fruit fly needs to make effective use of this information to successfully navigate the world.

A new section (outlined below in the “Changes Demonstrating Renderer Use”) has been introduced to give an example of the render’s usage.

Line 227: "Currently CompoundRay implements only a simple texture lookup, interpolating the nearest pixels of the texture associated with the geometry intersected." What would be gained by, or what is lost now relative to, implementing the full ray tracing approach?

The current approach allows arbitrary shaped eyes to be modelled and rendered in a fast way while maintaining the simplicity of the ray-casting approach, which is easier to expand in the future. In essence, compoundRay simulates pinhole-cameras as a regular polygon-projection approach would but doesn’t require abuse of the whole vertex-rendering pipeline to render these multiple cameras. Full ray-tracing could more accurately model light transport through the environments which we introduce as future work in the Discussion. Also see the section below “Changes Relating to Colour and Lighting” below.

Line 247: "Both pipelines share a common geometry acceleration structure and material shaders so as to save device memory." How is this possible? Can you hand-hold the reader a bit more to clarify how can both pipelines share the same GPU shaders? I may have missed a critical point.

This has now been explained (line 264).

Line 253: "rendering using a traditional camera skips steps 1", you mean, your software can alternatively use traditional cameras in place of modeled ommatidia? I am not sure I get it.

Yes, for users to interact with the 3D environment for debugging and analysis purposes CompoundRay allows use of normal cameras as well as ommnitidial-based renderings. We have added a clarification of this point to the text (line 254-255).

Line 293: "inhomogeneous ommatidial properties". An example you could cite is the eyes of the robber flies, Wardill et al. 2017 Curr Biol., where a frontal fovea with enlarged facets is reported.

Thanks for this suggestion. We have added references as requested (lines 298 and 490)

Figure 6: seems to imply that overlap is not desirable? Overlap could allow for all kinds of spatial computations, from supraresolution to contrast enhancement to motion detection.

The caption has been updated to specify that in particular, being able to leverage the most useful parts of homogeneous configurations (i.e. some overlap) is the benefit of a heterogeneous eye design.

Line 309: the statements would apply only in the absence of a nervous system that would make full use of the ommatidial signals.And, in general: aren't insect compound eyes known to present significant overlap in the field of view of each ommatidium relative to its neighbors? Figure 6c would then be the most realistic, if ignoring post-processing by the optic lobes?

Thank you for this feedback, and you are correct in pointing out the differences between the data shown and real insect eyes. However, Figures5 and 6 are intended not to show accurate compound eye models per se, but rather to show the different information that is provided by different geometries (Figure 5) and varying omnitidial properties (Figure 6). This analysis is intended to convince users of the necessity for this level of detail rather than speak directly to actual insect eyes. However, these are precisely the type of research questions that we have designed CompoundRay to be used to investigate. In particular, figure 6c is modelled as being able to leverage the potentially useful aspects of both the under- and over-sampled views in subfigures a and b, however the exact aspects that are leveraged are left as a simplified selection (just packing cones of vision) to better demonstrate the possibilities present within heterogeneous eye designs.

Figure 7: qualitatively, the top rendered scene looks quite similar throughout. I am not able to notice much difference between the left and right ends. Figure 7 needs work to better convey your message, and to better establish or highlight the vertical relationship between the images in the top and middle and the plot at the bottom.

The image is now a simple split between the 1 and 70-sampled images, with their exact locations on the sampling graph highlighted. This exemplifies the difference between the two and puts the graph into better context.

Line 357: "using a genetic algorithm to ﬁnd the location", where are the technical details of this approach? Parameters? Framework used, or code? Is it your own code?

We have removed the reference to a genetic algorithm from the text as any optimisation or brute-force search would allow this parameter to be optimised. We feel that our new description better conveys the task to the reader without drastically with specific tools. New text in lines (368-373)

Line 440: "On top of the potential biological insights CompoundRay may be able to offer", indeed, your paper would be so much stronger if you showed one such biological insight. It would then become a reference software for compound eye vision.

Indeed, we have added an entirely new example experiment which shows CompoundRay being used with modern deep-learning tools to investigate small object detection of different eye designs. We feel that this addresses the issue raised by the reviewer and enhances the paper. Thanks for the suggestion. (See the introduced new section outlined below in the “Changes Demonstrating Renderer Use”).

Changes Relating to Colour and Light Sampling

Previous discussion of colour and light sampling was over-complicated and the reasons for differing choices were not justified. We have changed this by removing broad discussion of light transport simulation and instead directly describing what CompoundRay does, particularly within the context of it essentially simulating multiple individual pinhole-cameras in parallel. The reasoning for the selection of ray-casting methods as a basis for the system are mentioned within the methods and discussions sections, and examples of further light and colour sampling methods that could be built on top of CompoundRay are placed in the “Future Extensions” section.

Changes Demonstrating Renderer Use

Following the requests for a specific demonstration of the renderer’s usage we have added an example study that comprises of a comparative analysis of the difference between a more complete model of an insect eye (that includes the surface shape), and two simplified derivatives based on the ideas of pairs of or singular panoramic image sampling methods. This study uses *Apis mellifera* visual systems as a base, as there already exists a collection of high-fidelity 3D scans of these eyes, one of which was chosen here.

The example study focuses on explaining to the reader the methodology behind using CompoundRay in to perform these comparisons and offers an example path to follow to perform analysis. Basic qualitative analysis is performed on the results, highlighting some surprising behaviours and points of interest.

[Editors’ note: further revisions were suggested prior to acceptance, as described below.]

The manuscript has been improved but there are some remaining issues that need to be addressed, as outlined below:Figure 10: the legend is far too short. Needs more detail on what each panel is, particularly for the lower panels. Also, where in the methods is the neural network described? Could be referenced here. Also, "network" is not quite the right shorthand to describe a neural network. Was it a CNN? Or generically an ANN? Or a perceptron?Line 504: where is the Materials and methods description for the "4-layer fully-connected neural network"?Line 532: "neural network was trained across 100 epochs three times", why? Why not 200 epochs, or 50, and 2 times, or 10? Why a batch size of 64? Also, these details are hardly worthy of opening the "Experimental results" section. Rather, they belong to the methods, alongside explanations and figures demonstrating why the set of parameters chosen are sensible and appropriate for the problem at hand.Line 565 reference to "Figure 10 bottom left" could really use the lettering of panels. In addition to a far lengthier and detailed explanation on what, exactly, is in that panel and how it was generated. And if these panels are as critical as they seem to be to evaluate the performance of the 3 eye models, then they ought to be bigger and far more richly annotated, to assist the reader in understanding what is being plotted.

The above comments are addressed together as they relate to the same section and figure. We have taken a number of steps. Firstly, we have separated the previous Figure 10 into 2 figures (Figures5 and 11). Figure 5 describes the experimental procedure only and has a much fuller caption as requested. This supports a new dedicated section in the Methods (Example Inter-Eye Comparison Method, diff pdf lines 280-354) that details the methodology, parameters used etc. Finally, Figure 11 now presents just the example results which links directly to the Results section – we think everything flows better now. Overall, we believe that the new text addresses all of the concerns raised. We appreciate the opportunity to make this clearer for readers.

We have made the main results panels larger and added some annotations to highlight the primary regions of space in which there is a difference in information content for the differing eye designs. However, as outlined above the main contribution of this example experiment is the demonstration of the utility of CompoundRay for modern data-intensive analysis, the pilot data plots are thus intended to inspire users and not define new findings per se.

Line 538, isn't the fact that a simple "single eye" design achieved a similar utility score concerning? Perhaps this reveals a disconnect between the test, which runs for 100 epochs (as plotted in Figure 10), and the real world, where an insect needs to take split-second decisions from the let go.

We did similarly note that there seems little variation in the eye designs and we are sure that this is not related to epoch size. We have added an additional explanation on epoch size selection (diff pdf lines 352-354), and have reframed the task to reflect the use of the neural network (diff pdf lines 280-354) which should clarify this for readers. Specifically, we are learning a close-to-optimal mapping between visual field and position over the course of a number of epochs (here 100), but at test, this mapping is simply used in a single-shot approximation of this mapping function, as is the case with all feedforward neural networks. Thus, the epoch size has no influence at the test time outside of reducing the noise present in the mapping of visual input to relative position.

Line 603: if you see opportunities for modeling the "recently reported" (2017) microsaccadic sampling in fruit flies, then why didn't you? Or is this the subject of a paper in preparation or under review elsewhere?

We have not yet modelled this work, nor is it submitted elsewhere. We see the primary contribution of the paper as a new method that will facilitate multiple new studies of various aspects of compound eyes. We intentionally focussed on the contributions of the methods to keep the focus of the paper clear. This reflects the guidance for Tools and Resources article types: “Tools and Resources articles do not have to report major new biological insights or mechanisms, but it must be clear that they will enable such advances to take place, for example, through exploratory or proof-of-concept experiments. Specifically, submissions will be assessed in terms of their potential to facilitate experiments that address problems that to date have been challenging or even intractable.” We feel that adding more analysis from the pilot study will adversely shift the focus away from Methods and muddy the clarity for the readers.

We note that we are now initiating follow-up work that will use CompoundRay in a far more detailed study of compound eye designs which will be the subject of a separate publication in the future focused on the new insights gained. Inclusion of this work in this manuscript would make it overly long and again deviate from the communication of the Method. Rather we included this text to highlight one of the interesting research topics that this tool could be used to study.

A question for the discussion: with its intrinsically distributed light sampling, compound eyes surely work not with frame-processing downstream circuits but rather event-based circuits, such as those of the dynamic vision sensor or silicon retina (Tobi Delbruck et al., various papers). Could you discuss the implications of asynchronous compound eye-based sensors, particularly when considering uneven ommatidial size which is not available in camera-based eyes such as in mammals or cephalopods? In a fast-moving animal such as a flying honeybee, asynchronous processing surely has advantages for motion detection and obstacle detection and avoidance, and scene recognition (i.e., flowers to land on), or better, to decode the waggle dance language of conspecifics.Considering event-based computing could make for a much nicer discussion than the caveats of anthropomorphised, frame-based visual computing, or at least complement it well, outlining the intrinsic advantages of compound eyes.

Thank you for pointing out this potential usage of CompoundRay. Indeed, we agree that CompoundRay could be an excellent input for a simulated DVS pipeline due to its speed and reconfigurability. We have now added an extra sentence which makes this potential usage explicit. This potential technological usage compliments well some of the biological research directions mentioned in the previous sentence. (diff pdf lines 626-630)